# Histones with an unconventional DNA-binding mode in vitro are major chromatin constituents in the bacterium *Bdellovibrio bacteriovorus*

Antoine Hocher [1,2,8,9] ✉, Shawn P. Laursen [3,8], Paul Radford[4], Jess Tyson [4], Carey Lambert [4], Kathryn M. Stevens[1,2], Alex Montoya[1,2], Pavel V. Shliaha[1,2], Mathieu Picardeau [5], R. Elizabeth Sockett [4], Karolin Luger [6,7,9] ✉ & Tobias Warnecke [1,2,9] ✉

Histone proteins bind DNA and organize the genomes of eukaryotes and most archaea, whereas bacteria rely on different nucleoid-associated proteins. Homology searches have detected putative histone-fold domains in a few bacteria, but whether these function like archaeal/eukaryotic histones is unknown. Here we report that histones are major chromatin components in the bacteria *Bdellovibrio bacteriovorus* and *Leptospira interrogans*. Patterns of sequence evolution suggest important roles for histones in additional bacterial clades. Crystal structures (<2.0 Å) of the *B. bacteriovorus* histone (Bd0055) dimer and the histone–DNA complex confirm conserved histone-fold topology but indicate a distinct DNA-binding mode. Unlike known histones in eukaryotes, archaea and viruses, Bd0055 binds DNA end-on, forming a sheath of dimers encasing straight DNA rather than wrapping DNA around their outer surface. Our results demonstrate that histones are present across the tree of life and highlight potential evolutionary innovation in how they associate with DNA.

Eukaryotic genomes are organized by nucleosomes which are composed of four core histones that assemble into an octamer to wrap DNA in two tight superhelical turns[1]. This arrangement of histones severely restricts access to the genome. As a result, eukaryotes rely on a complex system of access control to coordinate DNA-templated processes such as transcription, replication and DNA repair[2]. Histones are among the most conserved and abundant proteins across eukaryotes[3–5]. The 'histone fold' comprises three alpha helices (connected by two short

loops) that dimerize in a head-to-tail 'handshake motif'[6]. Additional structural elements, including divergent N-terminal tails, distinguish the four eukaryotic core histones from each other[7,8]. Deletion or depletion of individual histones leads to transcriptional dysregulation, cell cycle arrest and, ultimately, cell death[9–12].

Smaller histones, consisting only of the histone fold, are pervasive in archaea, where they can play a major role in chromatin organization together with other nucleoid-associated proteins (NAPs)[13–16]. In

[1]Medical Research Council London Institute of Medical Sciences, London, UK. [2]Institute of Clinical Sciences, Faculty of Medicine, Imperial College London, London, UK. [3]Department of Molecular, Cellular, and Developmental Biology, University of Colorado Boulder, Boulder, CO, USA. [4]School of Life Sciences, Medical School, Queen's Medical Centre, University of Nottingham, Nottingham, UK. [5]Institut Pasteur, Université Paris Cité, CNRS UMR 6047, Biology of Spirochetes Unit, Paris, France. [6]Department of Biochemistry, University of Colorado Boulder, Boulder, CO, USA. [7]Howard Hughes Medical Institute, Chevy Chase, MD, USA. [8]These authors contributed equally: Antoine Hocher, Shawn P. Laursen. [9]These authors jointly supervised this work: Antoine Hocher, Karolin Luger, Tobias Warnecke. ✉e-mail: a.hocher@lms.mrc.ac.uk; karolin.luger@colorado.edu; tobias.warnecke@lms.mrc.ac.uk

some archaea, histones are among the most abundant proteins in the cell[15]. In the model archaeon *Thermococcus kodakarensis*, histones are essential[17]. Similar to their eukaryotic counterparts, archaeal histone dimers bend DNA around their outer surface by contacting three consecutive minor grooves of DNA. Three independent DNA interaction interfaces are formed by main and side chains of paired L1-L2 loops and the central α1-α1 interface[7,18]. Whereas eukaryotic histones are obligate heterodimers, archaeal histones such as HTkA/B from *T. kodakarensis* and HMfA/B from *Methanothermus fervidus* (henceforth HMf/HTk histones) can form homo- or heterodimers[19,20]. These dimers then oligomerize into stacks of variable size that wrap DNA to form slinky-like 'hypernucleosomes'[18,21,22].

Bacteria use a variety of small, basic NAPs to organize their DNA. The most widespread NAP is HU, but other abundant, lineage-restricted proteins exist[23,24]. Deleting individual NAPs is often not lethal to bacterial cells. This is even true for NAPs that, like HU in *E. coli*, are major constituents of the nucleoid[25]. Histones are generally considered to be absent from bacteria. However, homology searches have identified proteins with putative histone-fold domains in an eclectic set of bacterial genomes[26]. The role(s) of these proteins in organizing bacterial chromatin has not been investigated. In this Article, we show that histone proteins are major nucleoid components in *Bdellovibrio bacteriovorus* and *Leptospira interrogans* and, using a combination of structural and molecular techniques, provide evidence for how they interact with DNA and might organize the *B. bacteriovorus* genome.

## Results

### Histone folds are common in several bacterial clades

We carried out a systematic homology search for histone-fold proteins in bacteria. Using a phylogenetically balanced database of 18,343 bacterial genomes, we found 416 proteins that contain a predicted histone-fold domain (Supplementary Table 1). Of the genomes, 1.86% encode at least one histone-fold domain, compared with 92.8% of genomes that encode HU. In agreement with previous work, we identified two major histone size classes, containing either a singlet or a doublet histone fold[26]. Both classes are typically devoid of other recognized domains (Fig. 1a and Supplementary Table 1). Like their archaeal homologues, most bacterial singlets also lack the long, disordered N-terminal tails that are characteristic of eukaryotic histones[16,27]. Amino acid conservation across the three domains of life is particularly high in the L2 loop (especially the RKTV motif, Fig. 1c). In archaeal and eukaryotic histones, this region contacts DNA as part of the L1-L2 binding motif. The highly conserved 'RD clamp' stabilizes this arrangement. Bacterial singlets are on average six residues shorter than HMf/HTk histones. This is mostly due to a shorter α2 helix, which is diagnostic of bacterial singlet histones (Fig. 1b,c). Several residues that are normally present in archaeal and eukaryotic histones, including the 'sprocket arginine'[28] R19 that is held in position by a hydrogen bond with T54, are not conserved in bacterial histones (Fig. 1c). The N terminus as a whole exhibits considerable divergence, is more hydrophobic in nature and includes a conserved serine-lysine motif (S9-K10) that is not found in archaea.

### *B. bacteriovorus* has a nucleoid-localized histone

The phyletic distribution of histones across the bacterial domain is patchy (Fig. 1d and Supplementary Table 1). In some instances, this might indicate assembly contaminants or recent/transient horizontal gene transfer. However, we identified clades where histone-fold proteins are present in several closely related sister lineages. Such phylogenetic persistence is particularly evident in the phylum Bdellovibrionota (Fig. 2a and Supplementary Fig. 1) and the order Leptospirales (Supplementary Fig. 2).

*B. bacteriovorus* HD100, the model organism of the Bdellovibrionota, is a bacterial predator with a biphasic, bacterially invasive life cycle[29]. Small, motile attack-phase cells breach, enter through and subsequently re-seal the outer membrane of Gram-negative prey bacteria (for example, *E. coli*). From the periplasm, *B. bacteriovorus* then consumes its prey by secreting proteases and nucleases into the prey cytoplasm, culminating in cycles of replication and coordinated non-binary division to yield new attack-phase cells, which are released from the husk of the ravaged prey following induced rupture of its cell wall and outer membrane[30,31].

*B. bacteriovorus* encodes two predicted singlet histones (Supplementary Table 1). Bd0055 is a bacterial singlet with a detectable homologue in 67% of Bdellovibrionota genomes (Figs. 1c and 2a, and Supplementary Fig. 1). It is highly conserved at the amino acid level (Supplementary Fig. 3). Bd3044 is a longer, less well-conserved protein that is present in only 37% of Bdellovibrionota genomes (Supplementary Figs. 1 and 3). Previous transcriptomic data from across different stages of the *B. bacteriovorus* life cycle indicate high expression of Bd0055, especially during active replication in the host (Supplementary Fig. 4). We therefore focus on Bd0055 as a candidate global organizer of the nucleoid.

We used quantitative label-free proteomics in attack-phase cells to confirm high abundance of Bd0055 at the protein level (Fig. 2b; ranked 12th out of 2,125 proteins quantified). Of note, protein abundance in attack-phase cells is better correlated with RNA abundance in growth phase, consistent with a time lag in protein production where transcript levels during growth phase foreshadow protein levels in attack-phase cells (Supplementary Fig. 4).

To investigate cellular localization of Bd0055 and to monitor expression throughout the *B. bacteriovorus* life cycle, we generated *B. bacteriovorus* HD100 strains in which the native copy of *bd0055* was replaced with a version that was C-terminally tagged with either mCherry or mCitrine. Both tagged strains exhibit no gross morphological defects and carry out predation efficiently. We detect strong fluorescence from tagged Bd0055 throughout the life cycle, including in free-swimming attack-phase cells and following invasion of the *E. coli* prey (Fig. 2c). Compared to a previously characterized mCherry-tagged control protein with known cytoplasmic localization[32,33], the fluorescent signal emanating from Bd0055-mCherry is absent from the cell poles, consistent with localization to the nucleoid as captured by Hoechst staining (Fig. 2d–f). In line with the absence of a signal peptide, we find no evidence for secretion into the *E. coli* prey, suggesting that Bd0055 is unlikely to be used for manipulation of prey chromatin.

The nucleoid-to-cell size ratio (also known as nucleocytoplasmic ratio) in *B. bacteriovorus* attack-phase cells is very large, making nucleoid co-localization less apparent than it would be in bacteria where the nucleoid occupies a smaller fraction of the cell. In addition, extreme nucleoid compaction in attack-phase cells has been shown to compromise free diffusion of some fluorescent proteins through the nucleoid[34]. As a consequence, much of the fluorescence signal might come from the more accessible nucleoid periphery, complicating precise delineation of its origin (cytoplasm versus nucleoid). We therefore pursued two complementary approaches to gather further evidence that Bd0055 co-localizes with the nucleoid in a cellular context.

First, we expressed a Bd0055-GFP fusion protein in *E. coli*, where the nucleoid is less compact and occupies a much smaller fraction of the cell. We observe clear co-localization of Bd0055-GFP with the nucleoid and little GFP signal from the surrounding cytoplasm (Supplementary Fig. 5).

Second, to obtain orthogonal in vivo confirmation that Bd0055 associates with the *B. bacteriovorus* nucleoid, we carried out sucrose gradient-based nucleoid enrichment experiments coupled to quantitative proteomics. We analysed two fractions from sucrose gradients of lysed attack-phase cells: a lower-density fraction that is enriched for soluble cytosolic proteins and a nucleoid fraction enriched for DNA and DNA-binding proteins. Subunits of the RNA polymerase are, as in other bacteria, very abundant (Fig. 2g). Bd0055, however, stands out as the most abundant protein in the nucleoid fraction, where it is

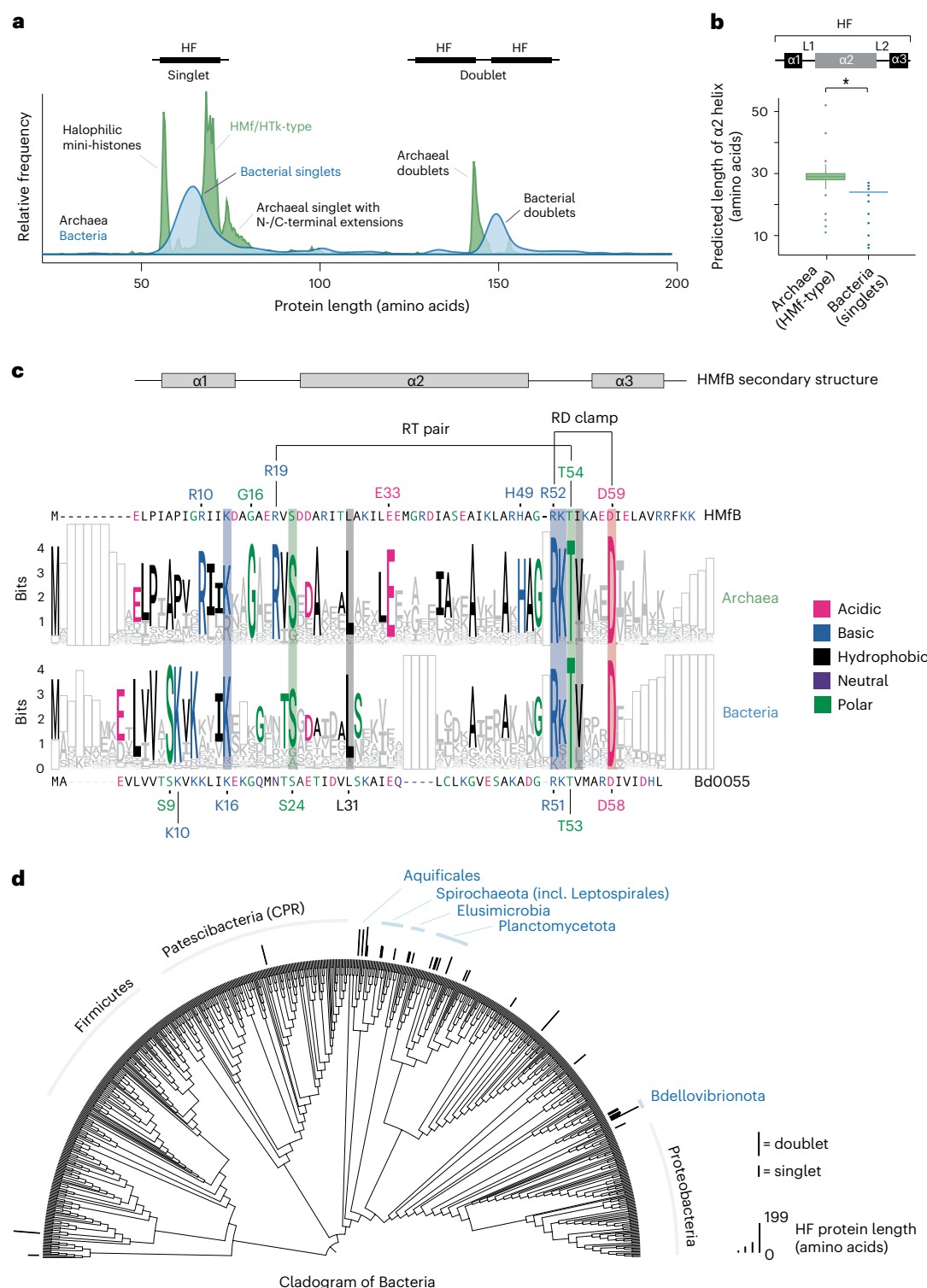

**Fig. 1 | Histone-fold proteins in bacteria. a**, Length distribution of proteins (<200 amino acids) encoding predicted histone-fold domains in bacteria and archaea. Relative frequencies are shown. **b**, Length of the α2 helix in bacterial versus HMf/HTk archaeal singlet histones (*, $P = 3.24 \times 10^{-185}$, two-sided Mood test, $N = 1,278$ archaeal histones, $N = 180$ bacterial histones; box plots show median and interquartile ranges (IQR), with whiskers extending to 1.5*IQR and values beyond this point plotted individually). **c**, WebLogo representation for bacterial versus archaeal singlet histones. Bacterial (bottom panel) and archaeal (top panel) histones were aligned separately, followed by profile–profile

alignment to allow comparison across kingdoms. Alignment gaps were coded as a separate character to retain their information value and are visualized as empty boxes. Residues that are notably conserved across kingdoms are highlighted by shaded boxes. **d**, Abridged (see Methods) bacterial species tree illustrating the phyletic distribution of histone-fold-containing proteins across the kingdom. Histones are represented as bars, the length of which is scaled to capture relative protein length in amino acids. Shorter bars indicate singlets and longer bars represent doublets or, on occasion, proteins with additional domains (see Supplementary Table 1 for details).

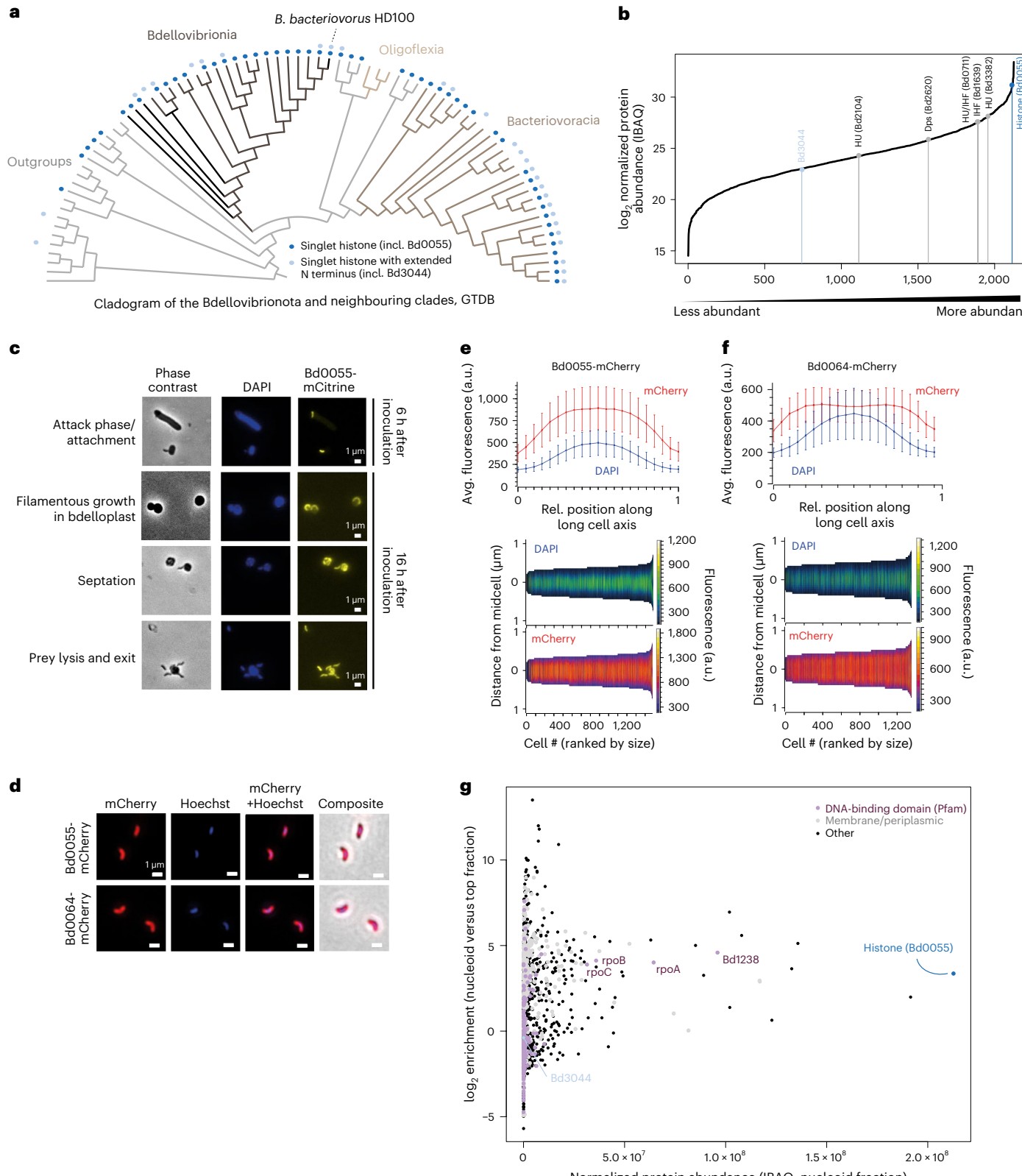

**Fig. 2 | Histones in *Bdellovibrio*. a**, Phyletic distribution of singlet histones across the Bdellovibrionota. **b**, Ranked protein abundance in *B. bacteriovorus* attack-phase cells, based on quantitative label-free proteomics. All quantified proteins are plotted. **c**, Representative images of different phases of the *B. bacteriovorus* life cycle from strains expressing a Bd0055-mCitrine fusion protein. **d**–**f**, Representative images and quantification describing the localization of Bd0055-mCherry and Bd0064-mCherry, a protein with previously established cytosolic localization, in *B. bacteriovorus*. Imaging experiments in **c**–**f** were carried out in triplicate. Number of cells used for quantification: *N* = 1,377 (Bd0055-mCherry); *N* = 574 (Bd0064-mCherry). Mean ± s.d. **g**, High abundance and prominent nucleoid enrichment of Bd0055 in *B. bacteriovorus* attack-phase cells (also see Supplementary Fig. 5). Note that the nucleoid fraction is also enriched for membrane components that have previously been reported to co-sediment with the nucleoid (see ref. 15 for previous work).

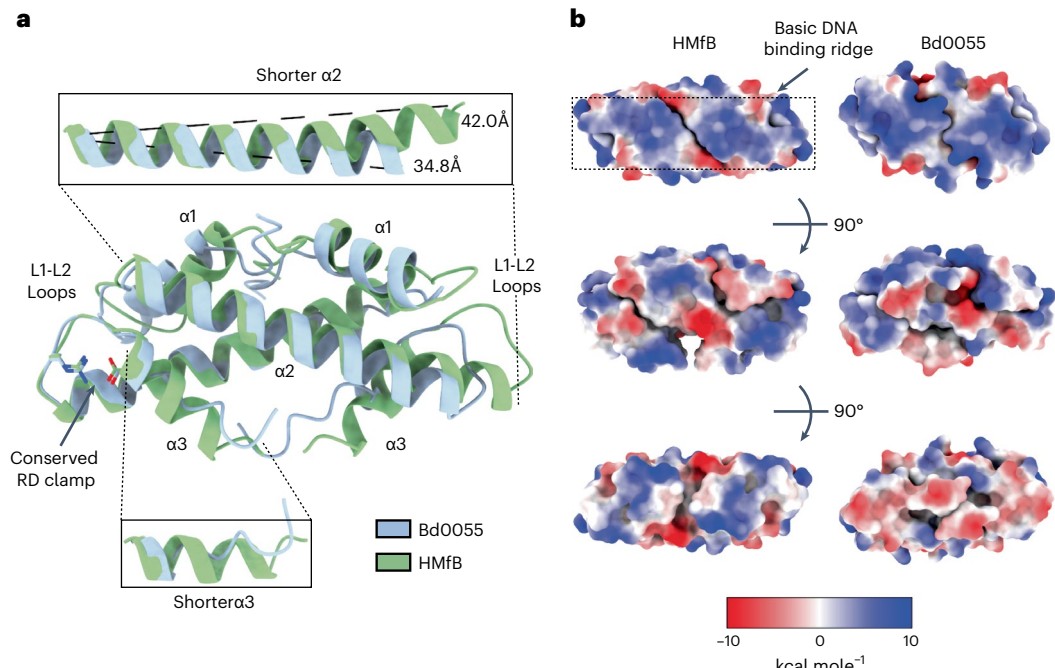

**Fig. 3 | Bd0055 forms a histone-fold dimer. a**, Crystal structure of Bd0055 superimposed onto archaeal histone HMfB (PDB 1A7W). Bd0055 maintains the overall topology of a histone fold, with shorter α2 and α3 helices. Bd0055 conserves the signature RD clamp and L1-L2 loops. **b**, Coulombic surface charge calculation of HMfB and Bd0055, shown in three orientations. Bd0055 maintains the basic ridge of residues important for binding DNA in other histones (top), but has a net acidic charge on its opposite face (bottom).

10-fold enriched compared with the top fraction. Homologues of classic bacterial NAPs encoded in the *B. bacteriovorus* genome, including HU and Dps, are present but less abundant and less enriched in the nucleoid fraction (Supplementary Fig. 5). The only other predicted DNA-binding protein that is notably abundant is Bd1238 (Fig. 2g), a yet uncharacterized HTH domain-containing protein with similarity to σ[54]-dependent transcriptional regulator proteins in other bacteria.

**Bd0055 folds into a histone-fold dimer**

We solved the crystal structure of Bd0055 to 1.8 Å resolution (Supplementary Table 2). Bd0055 forms a crystallographic dimer where one monomer is related to its partner through two-fold symmetry (Fig. 3a). It has the overall topology of a histone-fold dimer, as well as a positively charged ridge along its 'top' surface that is also found in archaeal and eukaryotic histones, where it is used for DNA binding (Fig. 3b)[1,18]. Of note, the Bd0055 α2 helix is 7 Å shorter than its archaeal and eukaryotic histone counterparts (1 helical turn shorter compared with archaeal HMfB, Fig. 1c), which decreases the overall size of the histone dimer (Fig. 3a). In addition, the α3 helix in Bd0055 forms only one helical turn and the remaining amino acids pack against α2 of the dimerization partner in an extended configuration. This organization differs from the three-turn α3 helix in archaeal/eukaryotic histones, which takes part in tetramerization by contributing to the four-helix bundle interface[18]. A conserved histidine in α2 (H49) is conspicuously absent from bacterial histones (Fig. 1c). Finally, the surface opposite the basic ridge of the Bd0055 dimer is more acidic than that of HMf/HTk histones (Fig. 3b).

**Bd0055 DNA-binding differs from that of archaeal histones**

Bd0055 interacts with DNA in vitro, as evident from fluorescence polarization (FP) assays (Fig. 4a). Using gel electromobility shift assays, we previously demonstrated that the archaeal histone HTkA shifts 147 bp DNA to a single discrete band[22]. HTkA saturates this DNA fragment at a ratio of ~10:1 and is unable to produce higher shifts with added

protein. In contrast, Bd0055 shifts DNA to several regularly spaced bands indicative of multiple binding events (Fig. 4b). Increasing the protein/DNA ratio of Bd0055 continues to decrease electrophoretic mobility, suggesting a different binding mode (Fig. 4b).

To test whether Bd0055 bends DNA into a nucleosome-like geometry, we assembled histones on 147 bp DNA fragments end-labelled with a Förster resonance energy transfer (FRET) donor-acceptor pair. Titration of archaeal HTkA onto this DNA brings the ends into FRET proximity by forming nucleosome-like structures, as observed previously[18]. In contrast, no signal was observed upon titrating even a large excess of Bd0055 (Fig. 4c). By monitoring fluorescence polarization of the same samples, we verified that both proteins bind DNA under these conditions (Fig. 4a; see Supplementary Fig. 6 for the effects of pH and salt on binding affinity).

To further investigate this different behaviour of Bd0055, we superimposed the Bd0055 histone structure onto an archaeal hypernucleosome consisting of four HMfB dimers and 118 bp of DNA (based on PDB 5T5K) and carried out all-atom molecular dynamics simulations. While archaeal histone dimers remain stably stacked throughout the simulation, the modelled Bd0055 hypernucleosome unfolds within a few nanoseconds (Fig. 4d and Supplementary Movie 1). Although Bd0055 dimers remain bound to DNA during the simulation, they no longer contact other dimers through protein–protein interactions, suggesting a failure to form stable tetramers on DNA.

This conclusion is further supported by comparative micrococcal nuclease digests of chromatin from *E. coli* strains that express high levels of either HMfA or Bd0055. Whereas digestion of chromatin from the HMfA-expressing strain results in a ladder of bands consistent with hypernucleosome formation, fragments of a defined size that would be indicative of hypernucleosome footprints are not observed in the Bd0055-expressing strain (Supplementary Fig. 7).

These differences were further explored in vitro by analytical ultracentrifugation (AUC), a first-principle approach to assess molecular weight and shape of a macromolecular assembly. A 147 bp DNA

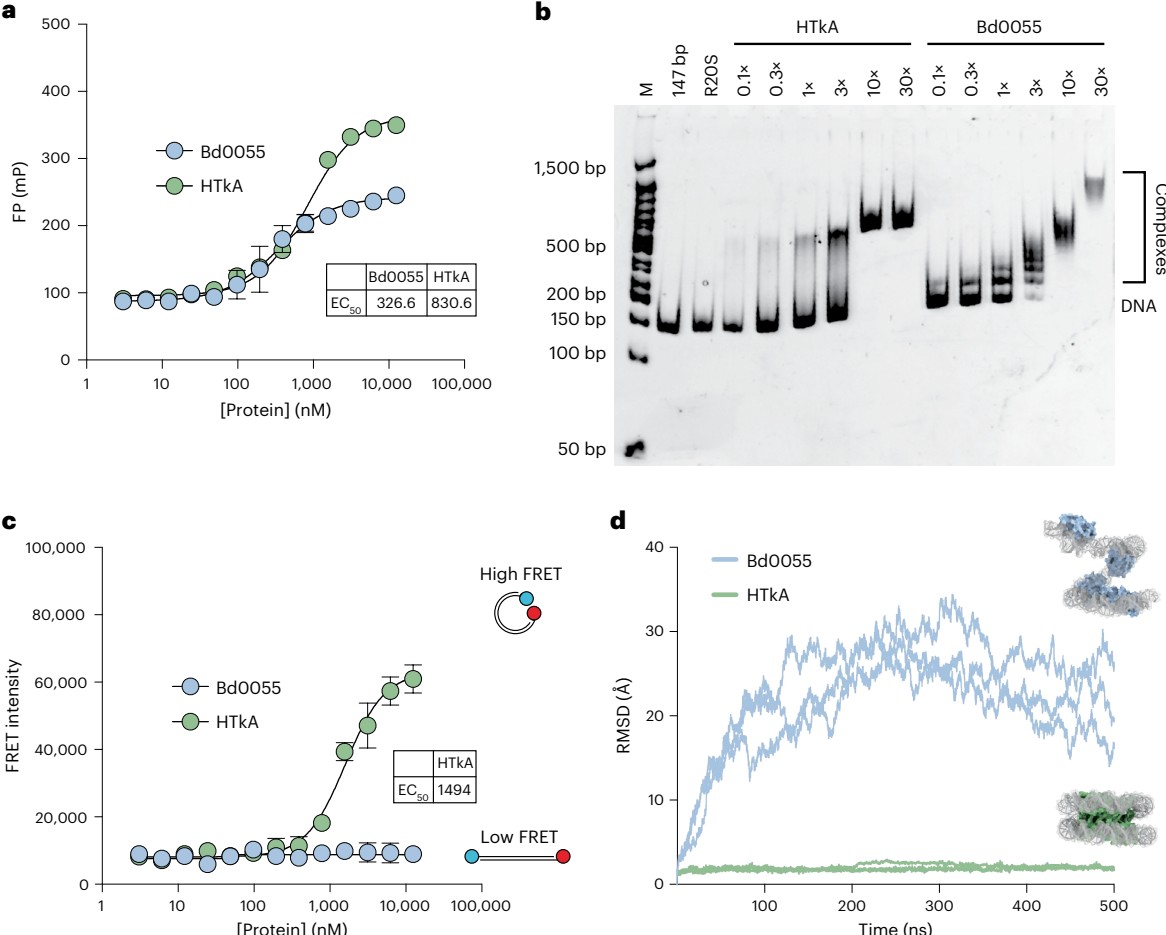

**Fig. 4 | Bd0055 binds DNA differently from HTkA. a**, Binding of Bd0055 to an Alexa488-labelled piece of 147 bp DNA (Widom 601 sequence), measured by FP ($N = 3$). EC$_{50}$, half-maximal effective concentration. Mean ± s.d. **b**, Bd0055–DNA binding monitored by EMSA (using unlabelled 147 bp DNA). Bd0055 binds with similar affinity as HTkA, but produces a different gel shift pattern. **c**, Bd0055 was titrated into dual-labelled 147 bp DNA (Alexa488 and Alexa647, same DNA as in FP experiment). At concentrations of HTkA high enough to bind DNA, histones wrap

the DNA and produce a high FRET signal. Bd0055 does not cause FRET between DNA ends, even with a large excess of protein ($N = 3$). Mean ± s.d. **d**, Plot of RMSDs from simulating a hypothetical Bd0055 hypernucleosome modelled onto PDB 5T5K using four histone dimers and 118 bp of DNA (see Supplementary Movie 1). After 500 ns of simulation, hypernucleosomes with HMfB remain stable, whereas hypothetical Bd0055 hypernucleosomes fall apart after as little as 10 ns.

fragment was saturated with 15 dimers of HTkA (S-value of 8 with no further increase), while over 60 dimers of Bd0055 can be added to the same DNA, resulting in a much larger complex that sediments at >12 S (Supplementary Fig. 8). This result provides orthogonal evidence for the very different binding mode and stoichiometry employed by HTkA and Bd0055.

To understand how Bd0055 binds DNA, we solved the crystal structure of Bd0055 in complex with a 35 bp DNA fragment (Supplementary Table 2). This structure shows that Bd0055 contacts the DNA with only one L1-L2 binding interface across the minor groove (Fig. 5a). The molecular details of this edge-on interaction are very similar to those observed for other known histones. It employs the L1-L2 motif that is conserved across the domains of life, although the sprocket arginine that reaches into the minor groove in nearly all other histone–DNA complexes is missing (Fig. 1c). Surprisingly, rather than the DNA bending around the positively charged histone surface, the DNA maintains a straight trajectory. Additional histone dimers bind the next two phosphates through L1-L2 interfaces engaged in identical interactions but flipped by 180 degrees (Fig. 5b). The filament is stabilized through protein–protein interactions between histone dimers that involve the basic DNA-binding ridge and the acidic underside of histone dimers 1 and 5, and through exchanging N-terminal tails between dimers 1 and 4

(Fig. 5c). This nucleohistone filament, which is characterized by a histone to DNA ratio of 1 histone dimer per 2.5 bp, reverses known histone convention by completely protecting straight DNA from the solvent. All other known histone–DNA complexes have histones on the inside and wrap DNA around them in a stoichiometry of 1 histone dimer per 30 bp, resulting in significant distortions of DNA double helix geometry (Fig. 5b).

**Proposed mechanisms for unusual Bd0055–DNA interactions**
Three main structural differences between Bd0055 and other histones could be responsible for this binding mode. First, the N terminus promotes interactions between dimers on the same strand of DNA, potentially stabilizing the fibre structure. Second, the shorter α2 helix of Bd0055 might force the DNA to bend too severely to wrap around the outside of the shortened dimer. Third, key residues responsible for tetramerization in archaeal histones are absent, most notably a histidine at the equivalent position of A48 (Figs. 1c and 5c).

To test whether any of these differences are responsible for the inability of Bd0055 to wrap DNA, we made the compensating mutants in vitro and assayed their ability to wrap 147 bp DNA using FRET. Neither deletion of the four N-terminal amino acids nor insertion of four amino acids into α2 of Bd0055 result in a protein that produces a FRET

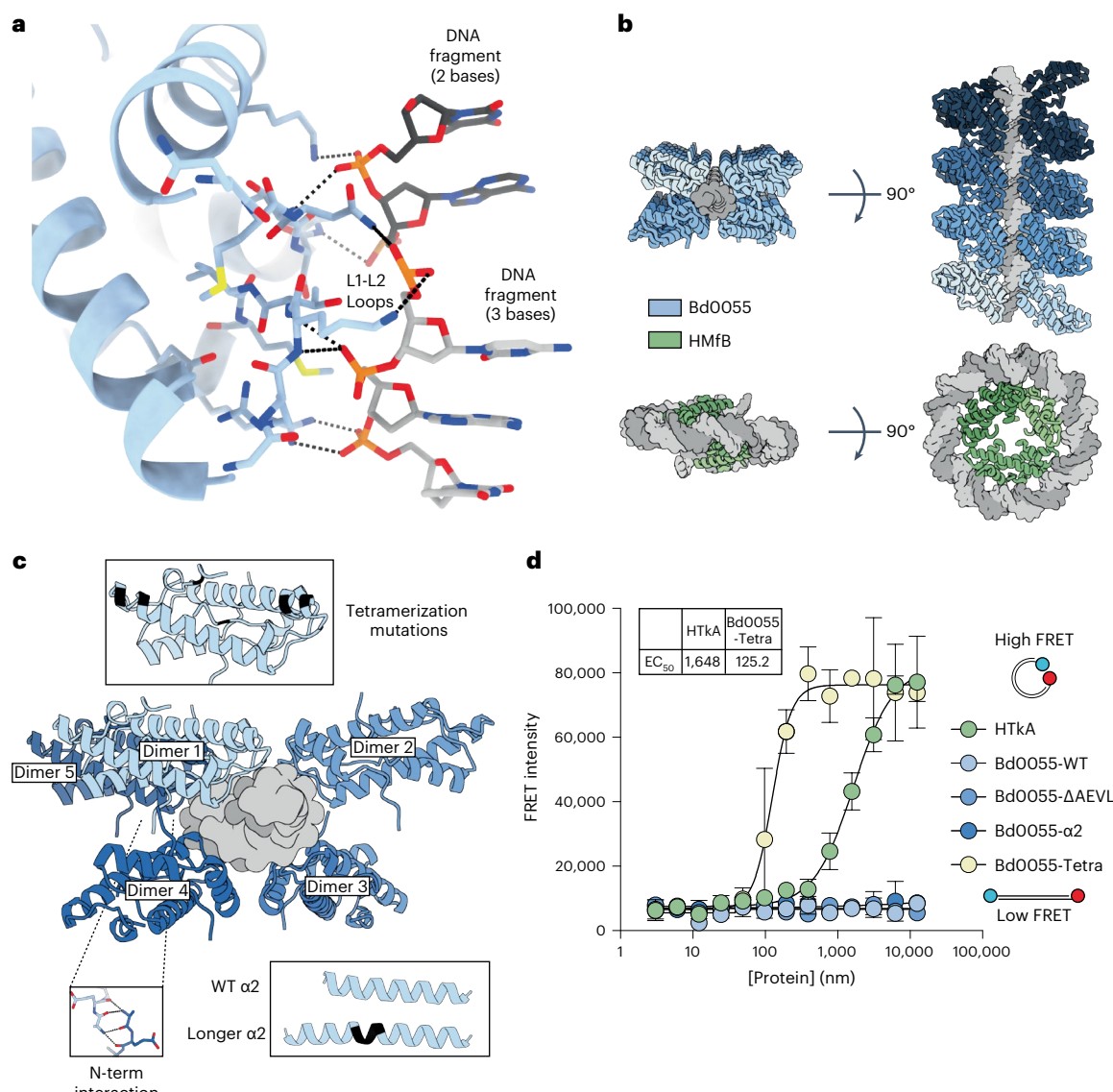

**Fig. 5 | Bd0055 binds DNA end-on and encases straight DNA. a**, Crystal structure of Bd0055 in complex with 35 bp DNA, showing interactions between the L1-L2 loop across and the phosphate backbone across the minor groove of DNA. Asymmetric unit is shown. **b**, Top: Bd0055 dimers encase dsDNA by binding to five phosphates (2–3 on each strand) and interacting with neighbouring dimers through electrostatic interactions. Consecutive dimers as they filament on the DNA are shown in decreasing shades of blue. Bottom: illustration of how archaeal HMfB wraps DNA around a core of histone dimers that are linked through a four-helix bundle structure (PDB 5T5K). **c**, Details of histone–histone interactions in the Bd0055 nucleohistone filament. Sites of mutagenesis to switch the binding mode are indicated. **d**, Mutation of A48H, S45F, I61L (Bd0055-tetra) in Bd0055 enables it to bring DNA ends into FRET proximity, while extending α2 by inserting YAIE (Bd0055-α2) or deleting the N terminus (Bd0055-ΔAEVL) has no effect, even though all mutants still bind DNA (see Supplementary Fig. 9). Mean ± s.d. of experiments carried out in triplicate.

signal between DNA ends (Fig. 5d). In contrast, mutations that mimic the archaeal tetramerization domain from HTkA (A48H, S45F, I61L; Fig. 5e) enable Bd0055 to bring the ends of the DNA within FRET distance. These amino acids are not implicated in DNA interactions in either binding mode nor does mutating them impair the ability of Bd0055 to bind DNA (Supplementary Fig. 9a). The magnitude of the FP signal observed for the tetramerization mutant binding 147 bp DNA closely matches that of HTkA and the shift to the left indicates the higher affinity of Bd0055 for DNA compared with HTkA (Fig. 5d). This particular mutant, when added to DNA, is more similar to HTkA than to wild type Bd0055 when analysed by AUC (Supplementary Fig. 8d), and structural prediction of the unsolved tetramer interface suggests that it may be capable of forming a similar interface to that in the crystal structure of HMfB (PDB 5T5K) and the predicted structure of

HTkA (Supplementary Fig. 9b–e). These data suggest that the inability of wild type Bd0055 to form tetramers is responsible for its inability to wrap DNA around its outer perimeter as is observed in archaeal and eukaryotic histones. This is consistent with previous findings from HMfB mutagenesis and analysis of a small set of archaeal histones that do not have a histidine residue at position 49. In either case, these histones were unable to form stable tetramer interfaces[20,35].

## Predatory and prey-independent growth probably require Bd0055

We sought to delete *bd0055* to investigate its physiological role. Unlike for *E. coli* and other model bacteria, advanced genetic tools to probe essentiality, such as inducible promoter systems, are not available for *B. bacteriovorus*. We therefore embarked on multiple attempts

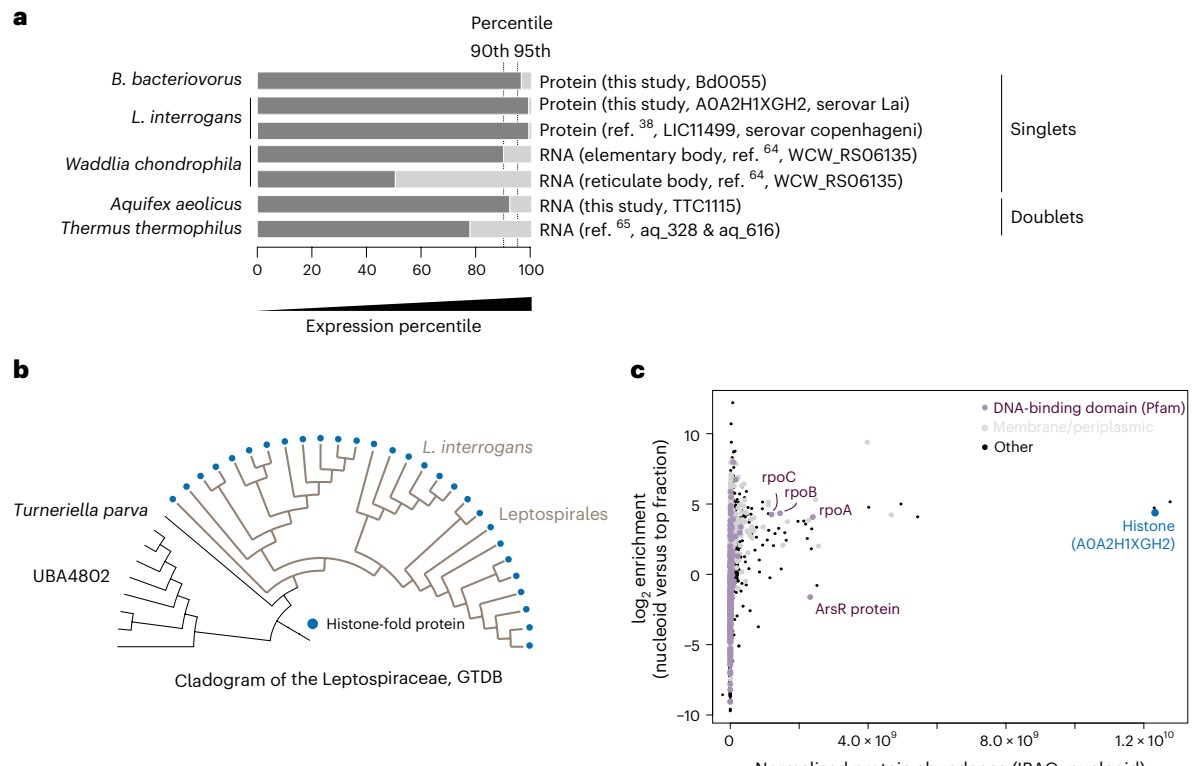

**Fig. 6 | Histones in other bacteria. a**, High expression of histone-fold proteins at the RNA and/or protein level is evident in bacteria from distant phylogenetic clades. Data sources: *L. interrogans*[38], *Waddlia chondrophila*[64],

*Thermus thermophilus*[65]. **b**, Phyletic distribution of singlet histones across the Leptospirales. **c**, High abundance and prominent nucleoid enrichment of the *L. interrogans* histone-fold protein.

to delete *bd0055* in the wild type HD100 background using an established silent gene deletion approach[31,36]. These attempts resulted in no deletion strains and 133 reversions to wild type. Reversions here are cases where growth occurred in selective conditions, suggestive of successful integration of the resistance cassette into the target gene but where subsequent examination revealed an intact copy of the target gene. This can occur, for example, as a result of merodiploidy, which is common in *B. bacteriovorus*. We also attempted to delete *bd0055* in a host-independent strain, HID13 (ref. 37), to establish whether impaired predation was the reason that no deletions were obtained in HD100. These attempts resulted in no successful deletions and 150 reversions to wild type. While not an ironclad demonstration of essentiality, the results above suggest that *bd0055* is probably important for fitness during both predatory and prey-independent growth. The results also imply that the second, less-abundant histone-fold protein, Bd3044, cannot readily compensate for the loss of Bd0055.

### Histones are abundant in bacteria outside Bdellovibrionota

Is *B. bacteriovorus* unique among bacteria in using histones as a major building block of chromatin? We examined publicly available gene expression data and generated additional transcriptomic and proteomic data. At both the transcript and protein levels, we find high expression of histone-fold proteins in several distantly related bacteria (Fig. 6a). We also observe notable phylogenetic persistence of histone genes in clades where gene expression profiles are not available, including in the Planctomycetota and Elusimicrobia (Supplementary Figs. 10–13).

We became particularly curious about *L. interrogans*, the causative agent of leptospirosis, whose histone (Uniprot ID: A0A2H1XGH2 gene ID: LA_2458) is exceptionally abundant. A previous study[38] found this histone to be the fourth most abundant protein in the cell (out of 1,502 proteins quantified), something we recapitulate, finding

it ranked second out of 2,433 proteins (Fig. 6a and Supplementary Fig. 14)[39]. Echoing results for Bd0055, this histone is highly conserved at the amino acid level, with homologues present in all members of the order Leptospirales, including saprophytic and pathogenic species (Fig. 6b, and Supplementary Figs. 2 and 15). Although the molecular details of its interaction with DNA remain to be elucidated, nucleoid enrichment experiments followed by mass spectrometry confirmed very high abundance and strong nucleoid enrichment (Fig. 6c). Finally, several attempts to delete the gene encoding this histone from the *L. interrogans* genome failed, suggesting that histones are probably also essential in this species.

## Discussion

Our results demonstrate that eukaryotes and archaea are not alone in using histones as major building blocks for chromatin. At least two bacteria, *B. bacteriovorus* and *L. interrogans*, also do so. The bacterial histone Bd0055 from *B. bacteriovorus* HD100 is structurally similar to its archaeal and eukaryotic homologues but, rather than wrapping DNA around its outer surface, binds DNA edge-on and oligomerizes to form a dense nucleohistone fibre that completely encases DNA in vitro. This is a different mechanism compared with that employed by known histones, which bend and distort DNA to form nucleosomes (in eukaryotes) or hypernucleosomes (in archaea), leaving DNA partially accessible. This inside-out arrangement presents a fundamental divergence from the accepted view of how histones structure DNA.

Whether, and under what conditions, such nucleohistone filaments form in vivo remains to be established, a challenge that is complicated by difficulties in generating deletion strains.

The life cycle of *B. bacteriovorus* may provide clues for how Bd0055 is used in vivo. Of particular relevance, attack-phase cells have highly compacted nucleoids that cannot be penetrated even by small fluorescent proteins[34]. The nucleoid decondenses shortly after entry into

host cells and a less compact nucleoid state persists until the *B. bacteriovorus* cell is ready to septate. At this time, the newly replicated genomes of each daughter cell again become discretely packaged into highly compacted nucleoids[39]. Bd0055 nucleohistone filaments might contribute to nascent genome segregation and/or the management of extreme compaction during attack phase. For example, at the nucleoid periphery, nucleohistone filaments could interface with and perhaps shield the remainder of the nucleoid from the cytoplasmic environment. Determining the genomic binding landscape of Bd0055, using chromatin immunoprecipitation followed by sequencing (ChIP-Seq) or related approaches, would be a logical next step to answer this question.

The crystal lattice we observed suggests the possibility that bacterial histones might generate higher-order fibre packing through the interaction of histones on adjacent nucleohistone filaments (Supplementary Movie 2). Although interactions between filaments do not seem to be very strong, it is worth pointing out that one of the three DNA-binding regions in the Bd0055 dimer is available for additional interactions with DNA, which might enable interaction between nucleohistone filaments. Tomograms from *B. bacteriovorus* and *L. interrogans* suggest occasional close packing of DNA fibres in vivo[40,41], but whether these are nucleohistone filaments remains to be established.

Histones probably also have a function in other bacteria. We present evidence for histone sequences in diverse organisms across the domain Bacteria. We find that the encoded proteins are often highly abundant and are likely to be essential in at least one other bacterium, *L. interrogans*. *Leptospira* histones share specific, important features with Bd0055, such as the shorter α2 helix, the conserved L2 loop and the absence of the histidine contributing to tetramerization in archaea. Our findings pave the way to investigating whether other bacterial histones bind DNA in the same way as Bd0055, when and how histone gene sequences were acquired and what function they serve. These may include functions outside of a DNA-binding context, as is the case for some eukaryotic histone folds that exist as part of macromolecular complexes such as TFIID, SAGA and CENP-TSWX (a component of the inner kinetochore complex), where they do not interact with DNA[42].

## Methods

### Homology survey of histone-fold proteins

PFAM HMM models of known histone-fold domains (Histone (PF00125), CBFD_NFYB_HMF (PF00808) and DUF1931 (PF09123)) were searched against a phylogenetically diverse database of 18,343 bacterial genomes (Supplementary Table 3) using hmmsearch from the HMMER package (v.3.1b2) with the −cut_ga option to ensure reproducibility. In addition, a list of eight prokaryotic histone-fold seed sequences were obtained from ref. [26] and used to direct homology search with Jackhmmer (HMMER v.3.1b2). Jackhmmer results with $P < 1 \times 10^{-3}$ were kept. Hits were combined and proteins longer than 200 amino acids discarded. All proteomes in the database were obtained from NCBI (accessed on 20 May 2020). When GenBank proteome files were not available, proteomes were predicted using Prodigal (v.2.6.3) with default settings.

For the analysis of α2 length (Fig. 1b), we considered curated sets of histone-fold proteins to facilitate the systematic identification of α2. Bacterial singlets were filtered from the bulk of histone-fold proteins by removal of DUF1931 HMM hits and further removal of proteins that were longer than 65 amino acids. HMf/HTk-type archaeal histones were pruned from a phylogenetic tree based on an alignment of bacterial and archaeal histone-fold proteins[20]. Secondary structures were then predicted for each histones using Jpred4 (ref. [43]) and the length of α2 calculated on the basis of these predictions. The α2 was taken to be the helix that overlapped the peptide region L28–L32 (in HMfB coordinates, see Fig. 1c) and was identified as the longest helix in the protein in 1,439 out of 1,458 cases. Outliers in α2 length (see Fig. 1b) were manually scrutinized and are owing to secondary structure misprediction, often splitting α2 up into two smaller helices. Removal of these outliers did not affect conclusions.

### Protein alignment and phylogenetic trees

All species trees were obtained from GTDB (https://gtdb.ecogenomic.org/). Trees were rendered by iTol (v.6.7) and finalized in Adobe Illustrator.

### Best reciprocal blasts

Best reciprocal blast hits used to compute amino acid conservation (Supplementary Figs. 3 and 15) were obtained using the blast_best_reciprocal_hit function from the R library metablastr (v.0.3), requiring a minimal *E*-value of 0.001.

### Bacterial culture

*B. bacteriovorus* HD100 was cultured on double-layer YPSC plates to initialize growth from frozen stocks and subsequently grown in liquid calcium/HEPES buffer containing *E. coli* S17-1 as described previously[44]. The kanamycin-resistant *E. coli* S17-1 (pZMR100) strain was used as prey for the culture of kanamycin-resistant *B. bacteriovorus* strains. Media were supplemented with kanamycin (50 µg ml⁻¹) when required.

The host-independent *B. bacteriovorus* strain HID13 was grown in YP medium at 30 °C as described previously[37].

*L. interrogans* was cultured at 30 °C in EMJH medium to a density of $10^7$ bacteria per ml. Cells were collected via centrifugation at $4,000 \times g$ for 20 min and the pellets were washed with phosphate buffered saline (PBS). The resulting pellets were frozen at −80 °C until use.

### Whole cell extract preparation

Total proteins were extracted from ~5 mg of −80 °C-frozen, unfractionated pellet using the iST proteomic kit (see below). Experiments were carried in biological duplicates at the protein extraction step and each biological replicate was itself technically duplicated at the proteomic step.

### Nucleoid enrichment and protein purification

Nucleoid enrichment was carried out as described in ref. [45] with minor modifications. Frozen pellets (~10 mg) were resuspended in 0.5 ml of buffer A (10 mM Tris-HCl pH 8, 5 mM EDTA, 100 mM NaCl, 20% sucrose) and 100 µl of buffer B (100 mM Tris-HCl pH 8.2, 50 mM EDTA, 0.6 mg ml⁻¹ lysozyme); after incubation on ice, 0.5 ml of buffer C (10 mM Tris-HCl pH 8.2, 10 mM EDTA, 10 mM spermidine, 1% Brij-58 and 0.4% deoxycholate) was added to the mixture. Lysozyme concentration in buffer B was doubled for *L. interrogans* (2× = 1.2 mg ml⁻¹); lysozyme incubation was carried out on ice for 10 min for *B. bacteriovorus* and at room temperature for 10 min for *L. interrogans*. Sucrose gradients (10–60%) were poured manually by 2 ml of 10% increment in ultra-clear (14 × 89 mm) Beckman–Coulter tubes. Gradients were allowed to cool down to 4 °C before the experiment. Lysed cells were deposited on gradients and centrifuged at 10,000 r.p.m. on a SW41-Ti Beckman Coulter rotor in a Beckman Optima centrifuge pre-cooled at 4 °C. Acceleration was set to the minimum value for both the start and the end of the run. Soluble cytosolic proteins settled at low density towards the top of the tube ('top fraction'), whereas the nucleoid, along with membrane proteins[46,47], settled at a higher density and could be identified as an opaque, viscous band ('nucleoid fraction'). Proteins from both fractions were concentrated and purified following a methanol chloroform treatment as described in ref. [15]. Samples were then prepared for mass spectrometry using the iST PreOmics kit, as described below. Experiments were carried out in biological triplicates at the protein extraction step and each biological replicate was itself technically duplicated at the proteomic step.

### Protein preparation for mass spectroscopy

Proteins from whole cell extracts and from different nucleoid enrichment fractions were all processed using the iST 8x Preomics kit. The sonication step was carried out as recommended by the manufacturer on a Bioruptor Plus (high-intensity setting). Following heat denaturation at 95 °C on an Eppendorf Thermomixer C and sonication, total protein amount was estimated at 205 nm absorbance using a NanoDrop

spectrophotometer (method scope 31). A total of 100 μg of material was subsequently used for whole cell extract and 37.5 (36.5) μg for *B. bacteriovorus* (*L. interrogans*) sucrose fractions. Enzymatic digestion (LysC/trypsin) was carried out on an Eppendorf Thermomixer C for 90 min at 37 °C at 500 r.p.m. For each experiment, all samples were processed using the same kit on the same day until storage at −80 °C in LC-load buffer.

### Liquid chromatography–tandem mass spectrometry (LC–MS/MS)

Chromatographic separation was performed using an Ultimate 3000 RSLC nano liquid chromatography system (Thermo Scientific) coupled to a Thermo Scientific LTQ Orbitrap Velos (*B. bacteriovorus* nucleoid enrichment samples) or an Orbitrap Q-Exactive mass spectrometer (all other proteomics samples) via an EASY-Spray source.

For sample analysis on the Velos, peptide solutions were injected and loaded onto a trapping column (Acclaim PepMap 100 C18, 100 μm × 2 cm) for desalting and concentration at 8 μl min⁻¹ in 2% acetonitrile and 0.1% trifluoroacetic acid. Peptides were then eluted on-line to an analytical column (EASY-Spray PepMap RSLC C18, 75 μm × 50 cm) at a flow rate of 250 nl min⁻¹. Peptides were separated using a 120 min stepped gradient, 1–22% of buffer B (i.e. 99-78% buffer A) for 90 min, followed by 22–42% buffer B for another 30 min (composition of buffer A: 95/5% H₂O/DMSO + 0.1% FA, buffer B: 75/20/5% acetonitril(MeCN)/H₂O/dimethyl sulfoxide (DMSO) + 0.1% formic acid (FA)) and subsequent column conditioning and equilibration. Eluted peptides were analysed on a mass spectrometer operating in positive polarity using a data-dependent acquisition mode. Ions for fragmentation were determined from an initial MS1 survey scan at 30,000 resolution, followed by collision-induced dissociation of the top 10 most abundant ions. MS1 and MS2 scan AGC targets were set to $1 \times 10^6$ and $3 \times 10^4$ for maximum injection times of 500 ms and 100 ms, respectively. A survey scan *m/z* range of 350–1,500 was used, with normalized collision energy set to 35%, charge state screening enabled with +1 charge states rejected and a minimal fragmentation trigger signal threshold of 500 counts.

For sample analysis on the Q-Exactive mass spectrometer, chromatographic separation was performed using an Ultimate 3000 RSLC nano liquid chromatography system (Thermo Scientific) coupled to an Orbitrap Q-Exactive mass spectrometer (Thermo Scientific) via an EASY-Spray source. Peptide solutions were injected and loaded onto a trapping column (Acclaim PepMap 100 C18, 100 μm × 2 cm) for desalting and concentration at 8 μl min⁻¹ in 2% acetonitrile and 0.1% TFA. Peptides were then eluted on-line to an analytical column (EASY-Spray PepMap RSLC C18, 75 μm × 75 cm) at a flow rate of 200 nl min⁻¹. See below for further details.

### Specific downstream settings for *B. bacteriovorus* whole cell extract proteomics

Peptides were separated using a 120 min gradient, 4–25% of buffer B for 90 min, followed by 25–45% buffer B for another 30 min (composition of buffer B: 80% acetonitrile, 0.1% FA) and subsequent column conditioning and equilibration. Eluted peptides were analysed by mass spectrometry in positive polarity using data-dependent acquisition mode. Ions for fragmentation were determined from an initial MS1 survey scan at 70,000 resolution, followed by higher-energy collision-induced dissociation (HCD) of the top 12 most abundant ions at 17,500 resolution. MS1 and MS2 scan AGC targets were set to $3 \times 10^6$ and $5 \times 10^4$ for maximum injection times of 50 ms and 50 ms, respectively. A survey scan *m/z* range of 400–1,800 was used, normalized collision energy set to 27 and charge exclusion enabled for unassigned and +1 ions. Dynamic exclusion was set to 45 s.

### Specific downstream settings for *L. interrogans* proteomics

Peptides were separated using a 90 min gradient, 4–25% of buffer B for 60 min, followed by 25–45% buffer B for another 30 min (composition of buffer B: 80% acetonitrile, 0.1% FA) and subsequent column conditioning and equilibration. Eluted peptides were analysed by mass spectrometry in positive polarity using data-dependent acquisition mode. Ions for fragmentation were determined from an initial MS1 survey scan at 70,000 resolution, followed by HCD of the top 10 most abundant ions at 17,500 resolution. MS1 and MS2 scan AGC targets were set to $3 \times 10^6$ and $5 \times 10^4$ for maximum injection times of 50 ms and 100 ms, respectively. A survey scan *m/z* range of 350–1,800 was used, normalized collision energy set to 27 and charge exclusion enabled for unassigned and +1 ions. Dynamic exclusion was set to 45 s.

### Proteomics data processing

Data were processed using the MaxQuant software platform (v.1.6.10.43)[48], with database searches carried out by the in-built Andromeda search engine against the GenBank proteome of each organism. A reverse decoy database approach was used at a 1% false discovery rate (FDR) for peptide spectrum matches. Search parameters were as follows: maximum missed cleavages set to 2, fixed modification of cysteine carbamidomethylation and variable modifications of methionine oxidation, protein N-terminal acetylation, asparagine deamidation and cyclization of glutamine to pyro-glutamate. Label-free quantification (LFQ) was enabled, with an LFQ minimum ratio count of 1. The 'match between runs' function was used, with match and alignment time limits of 0.7 and 20 min, respectively.

Data generated on the Orbitrap Q-Exactive were also processed using an alternative pipeline, pFind (v.3), which allows unbiased identification of peptide modifications present in a given sample[49]. The top five modifications identified by this pipeline were added to the MaxQuant search space and normalized abundance (IBAQ) computed as before. This added step did not affect normalized abundance estimates for our proteins of interest, and abundance values across the entire sample with or without this added step are very well correlated (rho > 0.99, $P = 0$).

### Nucleoid enrichment analysis

Relative nucleoid enrichment analysis was carried out[15] after quantification of proteins by MaxQuant. We compared the protein intensities (LFQ) from the soluble and the nucleoid-enriched fraction using the R package DEP (v.1.8.0). Globally, we observed robust enrichment of proteins with predicted membrane localization, providing validation for the approach (Supplementary Fig. 16).

### RNA extraction and sequencing

Total RNA was extracted from exponentially growing host-independent *B. bacteriovorus* HID13 and from −80 °C-frozen *Aquifex aeolicus* pellets (purchased from the Archaeenzentrum Regensburg, Germany) using the RNEasy kit (Qiagen), including DNase I treatment. RNA quality was assessed using an Agilent 2100 Bioanalyser.

For RNA extraction from *B. bacteriovorus* HID13, cells were diluted in 10 ml of fresh YP medium at an optical density (OD) of 0.1 and grown until reaching a density of 0.6 (20 h at 30 °C with shaking). For each replicate, 2 ml of culture were spun at maximum speed, snap frozen and kept at −80 °C until RNA extraction.

For the *B. bacteriovorus* samples (all RNA integrity number (RIN) scores >9.5), ribosomal RNA was depleted using the NEBNext rRNA depletion kit (Bacteria) and libraries made using the NEBNext Ultra II Directional RNA Library prep kit for Illumina according to manufacturer instructions. Paired-end 55 bp reads were generated on a NextSeq 2000 sequencing system with dual 8 bp indexing. For the *A. aeolicus* samples (all RIN scores >5), rRNA was depleted using an Illumina Ribozero kit (Bacteria) and libraries made using the TruSeq Stranded Total RNA LT kit according to manufacturer instructions. Single-end 50 bp reads were generated on a MiSeq with single 6 bp indexing.

### RNA-seq analysis

Untrimmed reads were mapped using Bowtie2 (v.2.4.4)[50] for *A. aeolicus* (single-end reads) and using BWA (v.0.7.17) for *B. bacteriovorus*

(paired-end). Read counts of all genes were estimated using the Python package HTSeq (v.0.6.1)[51].

## Fluorescence microscopy in *B. bacteriovorus*

The fluorophore mCherry/mCitrine was fused to the C terminus of Bd0055 by PCR amplification of the gene without its stop codon and amplification of the fluorophore gene. This was followed by Gibson cloning, using the Geneart assembly kit (Invitrogen) according to manufacturer instructions, into the mobilizable broad host range vector pK18*mobsacB*, and this was conjugated into *B. bacteriovorus* HD100 as described previously[52]. PCR amplification was carried out with phusion polymerase (New England Biolabs) according to manufacturer instructions (see Supplementary Table 3 for primers).

Cells were imaged on a Nikon Ti-E epifluorescence microscope equipped with an Apo ×100 Ph3 oil objective lens (NA: 1.45) and images were acquired on an Andor Neo sCMOS camera using Nikon NIS software. The following filters were used for fluorescence images: mCherry (excitation: 555 nm, emission: 620/60 nm), DAPI (Hoechst 33342): (excitation: 395 nm, emission: 435–485 nm). DNA was stained by the addition of Hoechst 33342 at a final concentration of 5 µg ml$^{-1}$. Images were analysed with FIJI software with the MicrobeJ plugin[53]. mCitrine-expressing cells were imaged on a Leica DMRB microscope with phase contrast and differential interference contrast for transmitted light illumination. To image mCitrine during predation, cells were fixed (1% paraformaldehyde, 5 min, quenched using 150 mM glycine) and DNA stained with DAPI in 1× PSB (5 min of stain removal by centrifugation).

## Fluorescence microscopy in *E. coli*

A vector allowing the expression of a Bd0055 C-terminal GFP fusion (separated by a GGSGGGGSGG flexible linker) was ordered from ATUM (backbone reference pD441-CC, T5 promoter). This vector was transformed into *E. coli* K-12 MG1655. To monitor the localization of Bd0055-GFP, freshly inoculated cultures (1:100, stationary phase inoculum) were grown for 1.5 h in LB medium, after which a final concentration of 1 mM isopropyl-β-D-thiogalactopyranoside (IPTG) was added to induce expression. Cells were collected 3 h after induction, fixed using paraformaldehyde (1% for 10 min at 37 °C) and quenched using glycine. The nucleoid was visualized using DAPI.

## Protein purification from *E. coli*

The Bd0055 (or mutant) open reading frame was codon optimized for expression in *E. coli* and synthesized into a double-stranded DNA (dsDNA) gBlock (IDT) with 35 bp and 22 bp overhangs on the 5' and 3' sides, respectively. The gBlock was cloned into a *lac*-inducible, ampicillin-resistant pET expression vector through restriction-free cloning. The protein was expressed in BD *E. coli* cells from overnight cultures using 0.4 mM IPTG induction after cells reached an OD of ~1.0. Cells were collected after 2 h and centrifuged at 6,000 r.p.m. for 20 min at 4 °C. Media were decanted and the cells were flash frozen in liquid nitrogen and stored at −80 °C. Cells were thawed on ice for 30 min and resuspended in lysis buffer (50 mM Tris pH 7.5, 5 mM EDTA, 0.1% Triton X-100, 5 mM b-mercaptoethanol (BME), 1 mM AEBSF protease inhibitor and 1 Pierce Complete protease inhibitor tablet per 50 ml). The cells were then lysed by sonication (3 rounds of 1 s on/off for 1 min) and centrifuged at 16,000 r.p.m. for 20 min at 4 °C. The lysate was filtered through a 0.45 µm filter and run over a 5 ml SP HP column (Cytiva) on an AKTA Pure FPLC using a linear gradient starting at buffer A (0 M NaCl, 50 mM Tris-HCl pH 7.5, 1 mM tris (2-carboxyethyl) phosphine (TCEP), 1 mM AEBSF) to 1 M NaCl (1 M NaCl, 50 mM Tris-HCl pH 7.5, 1 mM TCEP, 1 mM AEBSF) over 40 column volumes. Protein typically eluted around 300 mM NaCl. Fractions were sampled uniformly and analysed by SDS–PAGE, as Bd0055 has very little absorption at 280 nm. Fractions containing Bd0055 were pooled and diluted in buffer A to ~100 mM NaCl before being applied to a 5 ml Heparin HP column (Cytiva). Protein was eluted with a linear gradient from buffer A to buffer B over

40 column volumes (CV), with the protein typically eluting at 450 mM NaCl. Fractions were pooled and concentrated to a volume of 1 ml before being loaded onto a 120 ml S75 column (Cytiva) and run in buffer B containing 10% glycerol, with the protein having a typical retention volume of 74–82 ml. Fractions were pooled, concentrated, aliquoted, flash frozen in liquid nitrogen and stored at −80 °C.

## Apo protein crystallization

Recombinant Bd0055 in storage buffer (1 M NaCl, 1 mM EDTA, 50 mM HEPES pH 7.5, 10% glycerol) was crystallized through hanging drop diffusion in mother liquor (3.2 M (NH$_4$)$_2$SO$_4$, 100 mM BICINE pH 9) by gently mixing 1 µl of protein to 1 µl of mother liquor on a coverslip and sealing it onto a pre-greased well of a 24-well crystal tray containing 1 ml of mother liquor. Crystals formed within 24 h and matured within 3 d. Crystals were looped and frozen in liquid nitrogen. X-ray diffraction data were collected on a Rigaku XtaLAB MM003 X-ray diffractometer and indexed in DIALS (v.3.14). Molecular replacement was done using Phenix (v.1.20) and structures were refined in Coot (v.0.9.8.5). Short, idealized alpha helices were used as search models, and most of the remaining structure was built automatically using AutoBuild. Final residues were placed by hand in Coot.

## DNA binding (FP)

Protein (50 µM) was titrated using an Opentrons OT-2 liquid handling robot into 10 nM Alexa488 end-labelled 147 bp DNA by serial dilution down to a protein concentration less than the DNA probe in reaction buffer (10 mM NaCl, 1 mM EDTA, 50 mM HEPES pH 6). After incubation at r.t. for >15 min (overnight incubation did not change overall results), fluorescence polarization data were collected using a BMG Labtech CLARIOstar microplate reader. Data were analysed in GraphPad Prism (v.9) and fit to an [Inhibitor] versus response, four-parameter nonlinear regression curve.

## DNA binding (FRET)

FRET experiments were conducted simultaneously with the FP experiments, as the 147 bp DNA used was labelled on the opposite end of the Alexa488 with an Alexa647 fluorophore. Data were collected on a BMG Labtech CLARIOstar by excitation of 488 nm light while recording the emission at 647 nm. Data were analysed in Prism and fit to an [Inhibitor] versus response, four-parameter nonlinear regression curve.

Binding at different buffer conditions was determined via FP experiments using custom scripts for the OpenTrons OT-2 liquid handling robot, using 10 nM of 40 bp dsDNA Alexa488-labelled DNA (made by IDT) and screening four pHs (5, 6, 7 and 8) at seven different NaCl (0–250 mM) concentrations.

## Electrophoretic mobility shift assay (EMSA)

DNA (1 µM, 147 bp) was mixed with varying concentrations of protein in reaction buffer (10 mM NaCl, 1 mM EDTA, 50 mM HEPES pH 7.5). Reactions were mixed 1:1 with 80% glycerol, incubated at r.t. for >15 min and run on 10% native PAGE (0.2× TBE running buffer, 150 V, 90 min). Gels were stained with ethidium bromide and visualized on a Typhoon gel imaging system, using the appropriate laser and filter.

## MNase digestion

A codon-optimized, high-copy and rhamnose-inducible expression vector for Bd0055 was obtained from ATUM (pD861-SR). Inoculated from an overnight culture at 1:50, *E. coli* (strain W3110) was grown for 2 h in LB medium and for 4 additional hours in LB + 5 mM rhamnose. Mnase digestion was carried out as in ref. 54 with two modifications: culture volume was increased from 10 to 15 ml and digestion time was kept constant (15 min), while enzyme concentration was varied (Supplementary Fig. 7) with 1× corresponding to 4 U ml$^{-1}$ of Mnase (Thermo Fisher). Experiments were carried out in quadruplicate and a representative gel is shown in Supplementary Fig. 7.

## Molecular dynamics simulations

The HMfB hypernucleosome structure was built in ChimeraX (v 1.6) and Chimera (v 1.17.3)[55,56] by extending PDB 5T5K to include four HMfB dimers and 118 bp of dsDNA. The Bd0055 hypernucleosome structure was built by docking the Bd0055 apo structure into the HMfB hypernucleosome. All-atom molecular dynamics simulations using explicit solvent were carried out using AMBER18 using the ff14SB, bsc1 and tip3p forcefields (for protein, DNA and water, respectively). Structures were protonated and hydrogen mass repartitioned (as implemented through parmed in AMBER). Structures were placed in cubic boxes surrounding the structures by at least 25 Å, charge neutralized using potassium ions and hydrated with water molecules. The structures were energy minimized in two 5,000-step cycles, the first restraining the protein and DNA molecules to allow the solvent to relax and the second to allow the whole system to relax. Minimized structures were then heated to 300 K and slowly brought to 1.01325 atm. These systems were then simulated for 500 ns in 4 fs steps. Simulations were carried out on NVIDIA GPUs (RTX6000s or A100s) using CU Boulder's Blanca Condo cluster. Root mean square deviations (RMSD) analysis was carried out using cpptraj through AMBER18.

## DNA/protein crystallization

DNA fragments ranging between 35 bp and 45 bp in length, with randomized sequence and ~50% GC content, were screened for their ability to form discrete complexes with Bd0055, using EMSA. The 35 bp DNA (sequence: 5'TCTTGCACTAAGAGCTACTGGAGTGCGTCAGATGT3') was selected as it formed a discrete DNA shift at roughly 4:1 protein to DNA (it shifted to higher smears upon addition of more protein). DNA (75 μM) and 1,200 μM protein (1:16) were mixed in crystallization buffer (10 mM NaCl, 0.1 mM EDTA, 50 mM HEPES pH 7.5) and dialysed at r.t. against crystallization buffer. Hanging drop crystals were set in 4 μl drops at 1:1 complex to mother liquor (15% PEG 550 MME, 50 mM HEPES pH 8.0) and incubated at 20 °C. Crystals formed overnight and matured within a week. Large cubic crystals were looped, cryoprotected by washing in 30% glycerol and flash frozen in liquid nitrogen. X-ray diffraction data were collected at the ALS synchrotron (12,397.9 eV, 225 mm detector, $\Delta\Phi = 0.25$) and indexed in DIALS[57]. Molecular replacement was done using Phenix[58] and structures were refined in Coot[59] (as implemented through SBGrid[60]). The apo Bd0055 structure was used as a search model along with 2 single-stranded DNA (ssDNA) fragments (2 and 3 bp). The data indicated an asymmetric unit containing 2 ss nt + 3 ss nt (that are not paired), bound to one Bd0055 dimer. The high degree of symmetry in the crystal lattice is dictated by the protein, which overrules breaks in symmetry from DNA sequence.

## AUC

AUC experiments were carried out in a Beckman Coulter Proteomelab XL-A analytical ultracentrifuge using an An50TI rotor. Samples were prepared in 50 mM MES, 10 mM NaCl and 1 mM EDTA pH 6.0 using 500 nM 147 bp 601-sequence DNA (roughly 0.6 OD) and spun in a sapphire-windowed cell. Data were collected by monitoring absorbance during 150 scans at 30,000 r.p.m., 20 °C. Data were processed in UltraScan3, following standard processing protocols (2DSA, GA, GA-MC)[61]. Concentrations that crashed out of solution before first scans were acquired were not processed.

## Prediction of tetramer interfaces

Sequences for HTkA, Bd0055 or Bd0055-tetra were folded as tetramers using AlphaFold (v.2.3.2) in multimer mode[62]. Highest ranked models were then aligned to the same dimer of HMfB in PDB 5T5K using ChimeraX.

## Gene deletion

Attempts to generate a markerless deletion mutant of the *bd0055* open reading frame in *B. bacteriovorus* HD100 were by PCR amplification of the flanking 1,000 bp upstream region with the first 2 codons of the *bd0055* open reading frame and the final 3 codons with the downstream 1,000 bp flanking region and fusing these by Gibson assembly into the mobilizable broad host range vector pK18*mobsacB*. The primers used were: Bd0055_UP_F; Bd0055_UP_R; Bd0055_DN_F; Bd0055_DN_R (Supplementary Table 3).

This construct was conjugated into *B. bacteriovorus* HD100 as described previously[52] and resulting exconjugants screened for kanamycin sensitivity and screened by PCR to test for either gene deletion or revertant to wild type. Screening primers were: Bd0055KO_S_F 5' atctggagcttcacttcccg 3' and Bd0055KO_S_R 5' ggtgatgatccgggctctaa 3'.

For targeted mutagenesis in *L. interrogans*, a kanamycin resistance cassette replacing the coding sequence of LA_2458 and 0.8–0.9 kb sequences homologous to the sequences flanking the target gene was synthesized by GeneArt (Life Technologies) and cloned in an *E. coli* vector which is not able to replicate in *L. interrogans*. Plasmid DNA was then introduced in *L. interrogans* serovar Manilae by electroporation as previously described[63] with a Biorad Gene Pulser Xcell. Electroporated cells were plated on EMJH agar plates supplemented with 50 μg ml⁻¹ kanamycin. Plates were incubated for 4 weeks at 30 °C in sealed plastic bags or wrapped in foil to avoid desiccation.

## Reporting summary

Further information on research design is available in the Nature Portfolio Reporting Summary linked to this article.

## Data availability

All datasets are publicly available. Proteomics data have been deposited to PRIDE (PXD039405) and RNA-seq data to GEO under accession GSE220534. All genomes used were publicly available with no usage restriction (Supplementary Table 4). Structure data have been deposited at the Protein Data Bank (PDB 8FVX for apo Bd0055 and 8FW7 for DNA-bound Bd0055). Primers and oligos used in this study are provided in Supplementary Table 3. Raw microscopy images have been deposited in Zenodo (https://zenodo.org/record/8255694). Correspondence and requests for materials can be addressed to K.L. (karolin.luger@colorado.edu) or T.W. (tobias.warnecke@lms.mrc.ac.uk). Source data are provided with this paper.

## Code availability

Analysis scripts were run using R v.3.6.2. Scripts required to reproduce figures are available at https://github.com/hocherantoine/BHF/.

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

## Acknowledgements

We thank the LMS Genomics Facility for RNA sequencing, and A. Erbse at CU Boulder for help with crystallography. This study was funded by a Wellcome Trust Investigator Award in Science (209437/Z/17/Z, P.R., J.T., C.L., R.E.S.), a UKRI MRC core grant (MC-A658-5TY40, T.W., A.H., K.M.S.), and the Howard Hughes Medical Institute (K.L., S.P.L.).

## Author contributions

A.H. initiated the project and carried out all bioinformatic analyses, sequencing experiments and proteomics (including upstream bacterial culture and biochemistry). S.P.L. performed all in vitro work on *E. coli*-expressed Bd0055 (biophysical characterization, crystallography, structure analysis) as well as MD simulations. K.M.S. assisted in histone homologue curation and pan-bacterial analysis of histone proteins. A.M. and P.V.S. carried out proteomics work. P.R. built Bd0055-mCherry and -mCitrine fusion strains of *B. bacteriovorus* which were phenotyped microscopically by J.T. C.L. and P.R. attempted gene deletions of *bd0055* in predatory and prey-independent cultures. M.P. provided biomass for *L. interrogans* biochemistry and attempted gene deletions of LA_2458. R.E.S., K.L. and T.W. supervised the project. A.H., S.P.L., K.L. and T.W. led data analysis and interpretation, and wrote the paper with input from all authors.

## Competing interests

The authors declare no competing interests.

## Additional information

**Correspondence and requests for materials** should be addressed to Antoine Hocher, Karolin Luger or Tobias Warnecke.

# Reporting Summary

## Statistics

For all statistical analyses, confirm that the following items are present in the figure legend, table legend, main text, or Methods section.

| n/a | Confirmed | |
|---|---|---|
| ☐ | ☒ | The exact sample size (*n*) for each experimental group/condition, given as a discrete number and unit of measurement |
| ☐ | ☒ | A statement on whether measurements were taken from distinct samples or whether the same sample was measured repeatedly |
| ☐ | ☒ | The statistical test(s) used AND whether they are one- or two-sided *Only common tests should be described solely by name; describe more complex techniques in the Methods section.* |
| ☐ | ☒ | A description of all covariates tested |
| ☒ | ☐ | A description of any assumptions or corrections, such as tests of normality and adjustment for multiple comparisons |
| ☐ | ☒ | A full description of the statistical parameters including central tendency (e.g. means) or other basic estimates (e.g. regression coefficient) AND variation (e.g. standard deviation) or associated estimates of uncertainty (e.g. confidence intervals) |
| ☐ | ☒ | For null hypothesis testing, the test statistic (e.g. *F*, *t*, *r*) with confidence intervals, effect sizes, degrees of freedom and *P* value noted *Give P values as exact values whenever suitable.* |
| ☒ | ☐ | For Bayesian analysis, information on the choice of priors and Markov chain Monte Carlo settings |
| ☒ | ☐ | For hierarchical and complex designs, identification of the appropriate level for tests and full reporting of outcomes |
| ☐ | ☒ | Estimates of effect sizes (e.g. Cohen's *d*, Pearson's *r*), indicating how they were calculated |

*Our web collection on statistics for biologists contains articles on many of the points above.*

## Software and code

Policy information about availability of computer code

| Data collection | Crystallography: XDS, HKL3000; Rigaku XtaLAB MM003<br>Mass spectrometry : Ultimate 3000 RSLC nano liquid chromatography system (Thermo Scientific), LTQ Orbitrap Velos & Orbitrap Q-Exactive (Thermo Scientific)<br>Sequencing : MiSeq (Illumina); NextSeq 2000 (Illumina) |
|---|---|
| Data analysis | HMMER (v 3.1b2); Prodigal (v 2.6.3); Jpred4; iTol (v 6.7); R (libraries: metablastr (v 0.3), DEP (v 1.8.0)); MaxQuant (v 1.6.10.43); pFind (v 3); Bowtie2 (v 2.4.4); BWA (v 0.7.17); HTSeq (v 0.6.1); PHENIX (v 1.20); DIALS (v 3.14); Coot (v 0.9.8.5); ChimeraX (v 1.6); Chimera (v 1.17.3); Alphafold (v 2.3.2) |

For manuscripts utilizing custom algorithms or software that are central to the research but not yet described in published literature, software must be made available to editors and reviewers. We strongly encourage code deposition in a community repository (e.g. GitHub). See the Nature Portfolio guidelines for submitting code & software for further information.

## Data

Policy information about availability of data

All manuscripts must include a data availability statement. This statement should provide the following information, where applicable:

- Accession codes, unique identifiers, or web links for publicly available datasets
- A description of any restrictions on data availability
- For clinical datasets or third party data, please ensure that the statement adheres to our policy

All datasets are publicly available. Proteomics data have been deposited to PRIDE (PXD039405) and RNA-seq data to GEO under accession GSE220534. All genomes used were publicly available with no usage restriction (Table 53). Structure data have been deposited at the Protein Data Bank (PDB 8FVX and 8FW7).

## Research involving human participants, their data, or biological material

Policy information about studies with human participants or human data. See also policy information about sex, gender (identity/presentation), and sexual orientation and race, ethnicity and racism.

| | |
|---|---|
| Reporting on sex and gender | NA |
| Reporting on race, ethnicity, or other socially relevant groupings | NA |
| Population characteristics | NA |
| Recruitment | NA |
| Ethics oversight | NA |

Note that full information on the approval of the study protocol must also be provided in the manuscript.

# Field-specific reporting

Please select the one below that is the best fit for your research. If you are not sure, read the appropriate sections before making your selection.

☒ Life sciences       ☐ Behavioural & social sciences       ☐ Ecological, evolutionary & environmental sciences

For a reference copy of the document with all sections, see nature.com/documents/nr-reporting-summary-flat.pdf

# Life sciences study design

All studies must disclose on these points even when the disclosure is negative.

| | |
|---|---|
| Sample size | Sample size for biochemical/simulation experiments was n=3. Variance across experiments was low so explicit power calculations were not considered necessary. |
| Data exclusions | For FP and FRET experiments data was excluded at high protein concentration, where salt from the lM NaCl storage buffer had large effects (above 12.SuM protein) and below ~1nM protein, as all curves had flattened out by this point. |
| Replication | Experiments were replicated on different days (N=3 in every case). All replication attempts were successful. |
| Randomization | Random allocation with regard to covariates is not applicable to the experiments carried out. |
| Blinding | Blinding was not relevant for any of the experiments carrier out in this study. |

# Reporting for specific materials, systems and methods

We require information from authors about some types of materials, experimental systems and methods used in many studies. Here, indicate whether each material, system or method listed is relevant to your study. If you are not sure if a list item applies to your research, read the appropriate section before selecting a response.

## Materials & experimental systems

| n/a | Involved in the study |
|-----|----------------------|
| ☒ | ☐ Antibodies |
| ☒ | ☐ Eukaryotic cell lines |
| ☒ | ☐ Palaeontology and archaeology |
| ☒ | ☐ Animals and other organisms |
| ☒ | ☐ Clinical data |
| ☒ | ☐ Dual use research of concern |
| ☒ | ☐ Plants |

## Methods

| n/a | Involved in the study |
|-----|----------------------|
| ☒ | ☐ ChIP-seq |
| ☒ | ☐ Flow cytometry |
| ☒ | ☐ MRI-based neuroimaging |

