## [Peer Review File · Nature Microbiology]

Peer Review Information

Journal: Nature Microbiology

Manuscript Title: Histones with an unconventional DNA binding mode are major chromatin constituents in the bacterium *Bdellovibrio bacteriovorus*

Corresponding author name(s): Dr Tobias Warnecke

Editorial Notes:

This manuscript has been previously reviewed at another journal. This document only contains reviewer comments, rebuttal and decision letters for versions considered at Nature Microbiology. Mentions of prior referee reports have been redacted.

Reviewer Comments & Decisions:

Decision Letter, initial version:

Message: 16th June 2023

Dear Dr Warnecke,

Thank you for your patience while your manuscript "Histone-organized chromatin in bacteria" was under peer-review at Nature Microbiology. It has now been seen by 2 referees, whose expertise and comments you will find at the of this email. Now that I have read your revision plan, I am happy to proceed. For the parts of the rebuttal where any reviewer points to orthogonal methods to show essentiality please point out that the tools are unavailable in either *Bdellovibrio* or *Leptospira*. For the proteomics, please proceed as you have advised and include a line in the methods that explains the fixed/constant issue you write about.

I'm very sorry original reviewers 1,2 were unable to deliver reports, and will endeavour to keep your revision moving. As you pointed out, I usually suggest titles, so if you provide a set of options you are happy with this is always a massive help. Ideally what we are aiming for in any title is 10-15 words length, the main terms someone would use to search for the article being present, and the whole title being plain and informative.

We are very interested in the possibility of publishing your study in Nature Microbiology, but would like to consider your response to these concerns in the form of a revised manuscript before we make a final decision on publication.

If your paper is accepted for publication, we will edit your display items electronically so they conform to our house style and will reproduce clearly in print. If necessary, we will re-size figures to fit single or double column width. If your figures contain several parts, the parts should form a neat rectangle when assembled. Choosing the right electronic format at this stage will speed up the processing of your paper and give the best possible results in print. We would like the figures to be supplied as vector files - EPS, PDF, AI or postscript (PS) file formats (not raster or bitmap files), preferably generated with vector-graphics software (Adobe Illustrator for example). Please try to ensure that all figures are non-flattened and fully editable. All images should be at least 300 dpi resolution (when figures are scaled to approximately the size that they are to be printed at) and in RGB colour format. Please do not submit Jpeg or flattened TIFF files. Please see also 'Guidelines for Electronic Submission of Figures' at the end of this letter for further detail.

Figure legends must provide a brief description of the figure and the symbols used, within 350 words, including definitions of any error bars employed in the figures.

When submitting the revised version of your manuscript, please pay close attention to our [href="https://www.nature.com/nature-research/editorial-policies/image-integrity"](https://www.nature.com/nature-research/editorial-policies/image-integrity)>Digital

2Image Integrity Guidelines. and to the following points below:

Please include a statement before the acknowledgements naming the author to whom correspondence and requests for materials should be addressed.

Finally, we require authors to include a statement of their individual contributions to the paper -- such as experimental work, project planning, data analysis, etc. -- immediately after the acknowledgements. The statement should be short, and refer to authors by their initials. For details please see the Authorship section of our joint Editorial policies at http://www.nature.com/authors/editorial_policies/authorship.html

- * include a point-by-point response to any editorial suggestions and to our referees. Please include your response to the editorial suggestions in your cover letter, and please upload your response to the referees as a separate document.
- * ensure it complies with our format requirements for Letters as set out in our guide to authors at www.nature.com/nmicrobiol/info/gta/
- * state in a cover note the length of the text, methods and legends; the number of references; number and estimated final size of figures and tables
- * resubmit electronically if possible using the link below to access your home page: [redacted]
- * This url links to your confidential homepage and associated information about manuscripts you may have submitted or be reviewing for us. If you wish to forward this e-mail to co-authors, please delete this link to your homepage first.

Please ensure that all correspondence is marked with your Nature Microbiology reference number in the subject line.

Nature Microbiology is committed to improving transparency in authorship. As part of our efforts in this direction, we are now requesting that all authors identified as 'corresponding author' on published papers create and link their Open Researcher and Contributor Identifier (ORCID) with their account on the Manuscript Tracking System (MTS), prior to acceptance. This applies to primary research papers only. ORCID helps the scientific community achieve unambiguous attribution of all scholarly contributions. You can create

and link your ORCID from the home page of the MTS by clicking on 'Modify my Springer Nature account'. For more information please visit www.springernature.com/orcid.

We hope to receive your revised paper within three weeks. If you cannot send it within this time, please let us know.

[redacted]

Nature Microbiology

Reviewer Expertise:

Referee #1: Bacterial genetics, former reviewer 3
Referee #2: Bacterial chromosome organization

Reviewers Comments:

Reviewer #1 (Remarks to the Author):

The manuscript has undergone significant improvements in revision, particularly in terms of cohesion, coherence, and consistent data reporting, as well as the restoration of missing or mislabeled materials. In this round, I aim to provide a concise review focusing on two essential points to expedite the revision process and facilitate the decision regarding the publication of this report.

1. The discovery of the novel "insulator"-like structure of DNA-bound bacterial "histones" stands out as the strongest aspect of the report, both in terms of exploration and technical aspects.
2. However, the biological context of this discovery remains obscured by the sub-par whole-cell quantitative proteomics. In my previous review, I pointed out that the dimensionless "ranking" of protein amounts is meaningless, akin to recording the ranking of patients' temperatures in a hospital instead of their actual measurements. I also suggested several potential improvements for this experimental section. Unfortunately, the authors chose to dismiss this criticism by appealing to authority (Matthias Mann) and presenting arguments that seemed based on confusion or misunderstanding. It is evident that the authors were reluctant to rerun the proteomics experiments or reanalyze their data. This requires further clarification.

Regarding the mislabeled LFQ data graph, I initially indicated that a histone estimated at 14 molecules per cell (by Schmidt et al.) would only be sufficient to bind a negligibly small ~20 bp of DNA. However, the revised labeling now suggests an estimate of ~16,000 molecules per cell, which should be adequate to bind approximately 25 kb of genomic DNA, accounting for 1.5% of the total genome size of ~3.7 Mb. Unfortunately, this estimate is derived from a 2011 paper by different authors studying a different bacterium,

4and the current authors appear unwilling or unable to replicate these findings. Nevertheless, it is crucial to have absolute quantification of the protein to contextualize the intriguing histone-DNA complex and support the notion that it is a significant structural factor rather than a mere curiosity.

Moving forward, I see three options: 1) leave the issue of quantitative proteomics in its current unsatisfactory and uninformative state; 2) rerun the experiments by spiking the cell material with a known amount of recombinant protein (or a set of cognate synthetic peptides) to obtain "true" absolute quantitation; or 3) reanalyze the already generated data with additional sophistication that the authors previously deemed unnecessary. If the authors choose the third option, I have some encouraging news for them: upon spot checking the data made available in the second round, it was revealed that Bb0055 is not ranked #4 by mass—it is more likely around #15 or 20. However, in terms of molar amount (given its small size), it could very well be the second most abundant protein in the ranking of molecules per cell, possibly even surpassing the estimated 16,000 per cell.

A few points on proteomics analysis: citing Matthias Mann as an "undisputed leader in the proteomics field" does not sufficiently address the caveat that the method he chose for the coarse-grained, robotics-assisted, simultaneous analysis of 100 proteomes may not be the most suitable for fine-grained quantification of a few samples derived from a single bacterial proteome, with a specific emphasis on a small protein (64 amino acids) that is particularly sensitive to experimental setup idiosyncrasies.

Furthermore, my comment on the arbitrary assignment of constant and variable modifications was misconstrued by the authors as a query regarding post-translational modifications (PTMs), whereas it was actually about the impact of arbitrarily assigned modifications introduced during sample preparation and analysis, as opposed to modifications actually detected in a given sample using an open search tool. One such popular tool is pFind (<https://www.nature.com/articles/nbt.4236>), incidentally also used in Matthias Mann's paper (Extended data, Fig 10). In practical terms, using any modification as "constant" is appropriate only when it occurs in 100% of cases. Conversely, if it is significantly less than 100% in the samples under review (such as Cys→Carbamidomethyl(Cys)), it leads to a significant undercounting of this protein. Additionally, the arbitrary inclusion of "variable" modifications not present in sampled peptides in significant amounts drives up the false discovery rate (FDR), resulting in even more false negatives. In this particular case, most samples did not warrant the inclusion of Gln→Pyro(Glu), but instead required additional variable modifications (their omission also leads to false negatives).

This is why using open search tools such as pFind is important, not only for the sake of overall experimental realism but also for more accurate quantitation of small proteins. These tools generate fewer peptides, which, in turn, greatly amplifies the errors of coarse-grained analysis.

I acknowledge that rerunning the experiments entails additional costs and time expenditures. Therefore, I would accept a re-analysis of the existing data with improved experimental realism (as described above). A new version of MaxQuant has been released, which performs analyses that used to take days in a matter of minutes. I cannot accept "ranking" as a meaningful proteomics metric suitable for any reader, especially a general one. However, a dimensionless metric of normalized intensity (total intensity of protein-derived peptides normalized to its length) seems appropriate.

Reviewer #3 (Remarks to the Author):

Review for Histone-organized chromatin in bacteria

In this manuscript, Hocher et al., probe the mechanism of action of histone-like proteins in bacteria. Following from a previous study reporting the presence of proteins carrying histone folds in bacteria, authors in the present study first conduct a comprehensive search of histone-like proteins in bacteria and identify two classes of the same: singlet and doublet histones. They choose Bd0055 from *B. bacteriovorus* as their example of a singlet histone and go on to characterize the DNA binding mechanism of the same. They find that Bd0055 pulls-down with the chromatin fraction, is abundantly expressed, and binds DNA *in vitro*. Structural analysis reveals that Bd0055 is distinct from other reported histones as it does not tetramerize or bend DNA. Instead Bd0055 binds DNA ends-on, likely forming a nucleoprotein filament. Authors show that the lack of tetramerization is due to a missing c-terminal extension, that is found in eukaryotic and archaeal histones. Adding the same to Bd0055 results in tetramerization and DNA bending. They identify Bd0055-like proteins in other bacteria and show that this protein too is abundantly expressed and is likely associated with the nucleoid. They conclude that histone-like proteins are conserved in bacteria as well, although the mode of action is distinct.

The findings presented in this study are very interesting and have broad implications for our understanding of chromosome organization in bacteria, as well as across domains of life. Thus, overall, I am supportive of publication of this work in *Nature Microbiology*. The experiments are well-designed and clear and the data solidly support the proposed mechanism of action of Bd0055. The authors have also added significant additional experiments to address previous reviewer comments.

However, as with the previous review of this work, I too am of the opinion that the authors do not provide clear evidence for the possibility that these proteins are indeed functioning as 'histones' in organizing the bacterial chromatin. I think much of this can be resolved via modifications to some sections of the text (including title). In addition, the authors need to provide further evidence to support the possibility of chromatin-associated function for these proteins:

1. The title of the manuscript is misleading. The authors have not provided evidence that Bd0055 organizes the bacterial chromatin. *In vitro* evidence supports the idea that this protein might form a nucleoprotein filament. However, it is unclear whether this function has any consequence on chromosome organization or condensation *in vitro* or *in vivo*. In addition, these proteins appear to be sparsely conserved in bacteria. The authors should modify the title, and tone down sections of the abstract and discussion where they suggest broadly the conservation of histones (and their role in chromatin organization) in all domains of life.

2a. Lack of success in generating a deletion cannot be reported as the only support for the possibility that Bd0055 is essential. The authors dedicate an entire section of their results to describe the efforts to make a deletion of Bd0055. While I appreciate that this likely suggests that this gene is essential, I do not think a main result can be drawn from this observation unless other experiments are conducted. Hence, this section should be removed and perhaps one sentence can be included in the manuscript, where the authors

6indicate the inability to generate a deletion. To be able to conclude on essentiality of Bd0055, authors should express the gene from an inducible promoter to show growth defect when inducer is absent.

2b. As in the above comment, authors conclude essentiality of Bd0055-like proteins in other bacteria as well, but they do not provide sufficient data to support this conclusion. Hence the section has to be significantly re-written to remove reference to essentiality. To support the conclusion they are making in the last results section, authors should perform the experiment as described in comment 2a.

3. To support the conclusion that Bd0055 interacts with the bacterial chromosome, authors can carry out DAPI/ Bd0055 co-localization experiments in anucleated or elongated cells as well. For example, when expressed in *E. coli*, does Bd0055 associated with the nucleoid? In elongated *E. coli* cells (such as those blocked for cell division), the cytoplasmic space increases but the nucleoids remain compact. In this condition, does Bd0055 still localize with the nucleoids?

4. The MNase assay (Fig. S7) is important and should be included in the main figure. quantify the DNA-protection conferred by Bd0055 to MNase treatment.

Author Rebuttal to Initial comments

Referee #1:

Hocher et al., report their findings on the role of histone-like protein Bd0055 from B. bacteriovorus in organizing bacterial genome. They report that Bd0055 is expressed in vivo in high amounts 2) is a nucleoid-associated protein that co-purifies with the nucleoid, 3) interacts with DNA in vitro, 4) determine its molecular interactions with a short dsDNA strand. Based on these experiments, they propose that this histone-like protein binds to DNA and forms a nucleofilament.

While I appreciate that the findings are very intriguing and of potentially high interest, the current work does not provide the validation of their model, which is whether Bd0055 interacts with DNA and forms a functionally important protein:DNA structure in vivo. These key points must be addressed experimentally before this work can be considered for publication.

Here are my major points:

1. Authors should perform experiment/s that prove that show that Bd0055 organizes DNA in vivo or acknowledge that the conclusions of the paper regarding in vivo roles are in part based on previous tomographic work (Butan et al., 2011, Raddi et al., 2012).

7We cited both papers (Butan *et al.* and Raddi *et al.*) in the manuscript (page 10, 3rd paragraph) and explicitly pointed to the tomographic work the reviewer mentions. Unfortunately, there was a formatting error such that the references did not appear in the bibliography. We have rectified this mistake.

We consider demonstration of how Bd0055 organizes DNA *in vivo* outside of the scope of the current manuscript given the extensive additional work required, which is further complicated by the fact that Bd0055 is essential. High-end tomography is still not mainstream, and, unlike membranes or ribosomes, bacterial nucleoids are notoriously difficult to visualize. We have, however, carried out additional *in vitro* work looking at the behaviour of Bd0055 in solution. These new experiments support our conclusion that fundamental differences exist in how model archaeal histones and Bd0055 associate with DNA. These experiments are detailed below.

2. Can authors use a similar line of experimentation (e.g. MNase or DNase I-mediated degradation) that they used in their paper (Rojec *et al.*, 2019 PMID: 31692448)?

We carried out MNase digestion experiment as suggested. As previously done for the archaeal histones HMfA and HMfB in Rojec *et al.*, we heterologously expressed Bd0055 in *E. coli*. In parallel, we expressed HMfA from *Methanothermobacter fervidus* (as used in Rojec *et al.*) as a positive control. MNase digestion of chromatin from the Bd0055-expressing recombinant strain does not yield protected DNA fragments of a defined size. This is in stark contrast to digestion of chromatin from *E. coli* expressing HMfA, where a ladder of defined fragment sizes reflects the presence of (hyper)nucleosomal complexes of different length (Figure A below and new Figure S7). This further supports our findings from crystallography, FRET, and AUC (new data below), that Bd0055 does not form standard (hyper)nucleosomal particles. On a technical point, note that areas of strong signal on the Bd0055 gel could be misread to imply the existence of a relatively defined fragment size class.

This is not the case and rather reflects the fact that genomic DNA is quickly digested to low molecular weight species that accumulate at the bottom of the gel. This is evident from the fact that we observe a similar band pattern when digesting chromatin from the same strain but where Bd0055 was not induced (and is in fact actively tightly repressed by the addition of glucose, see Rojec *et al.* 2019, Figure A panel b). Finally, note that – while fragments of a defined size are absent – the digest (comparing induced and uninduced) does suggest somewhat greater protection (visible as smears centred at higher molecular weight in the induced condition) when Bd0055 is present, although the effect appears minor.

Figure A. MNase digestion of chromatin from *E. coli* strains heterologously expressing either the archaeal histone HMfA or the bacterial histone Bd0055.

3. The paper should be more specific about the histones that they are studying because this paper focused on singlet histones (particularly from *Bdellovibrionota*), but the title is generalized to all bacterial histones. In addition, the title talks about the function of “histone-organized chromatin in bacteria.” They assume that the role is to organize chromatin (based on the title), while they explicitly say in the paper that the function is still unknown.

The reviewer is correct in that we focussed our experimental efforts on *B. bacteriovorus* and, to a lesser extent, *Leptospira interrogans*. At the same time, in establishing evolutionary histories and mining public gene expression data (see Figures 1,5), the manuscript considers bacterial histones much more broadly. The current title seeks to reflect the most intriguing aspects of our work: that histones exist in the bacterial kingdom, and that they have a role in organizing chromatin. The title

does not suggest that all bacterial histones behave in the same way, nor does it assert function. While we lack knowledge of the specific functional roles of histones (e.g. in transcription regulation, defense against mobile elements, etc.), we provide ample evidence, for *Bdellovibrio* and *Leptospira*, that histones are a major constituent of chromatin in these species and must by that very fact affect organization of chromatin in these bacteria.

4. *They should acknowledge previous publications of similar structures of bacterial histone-like proteins (Qui et al., 2006 PMID: 16287087).*We added this citation in the first paragraph of the Results section. Note that this publication reports on the crystal structure of the doublet histone from *Aquifex aeolicus*, without further investigation into DNA binding or other in vitro/in vivo behaviour of this protein.

5. In the in vivo experiments, when Bd0055 is fused with a fluorescent protein, it is unclear if it co-localizes with the DNA, since it appears dispersed (Fig. 2c and 2d). Including an experiment using one of the several new technologies in fluorescence microscopy that can help improve the localization (e.g. super-resolution microscopy) would be appropriate. Otherwise, could the distribution of the Bd0055-mCitrine in the non-attack phases or Bd0055-mCherry, indicate that maybe organizing DNA is not the primary function of the protein?

We use the fluorescently tagged Bd0055 principally to monitor expression levels through the life cycle and to show that localization is consistent with nucleoid localization. We did consider super-resolution imaging to further probe nucleoid localization, but decided that any results here would have been hard to interpret. We know from prior work that the nucleoid in attack phase cells (on which we focus) is highly condensed and cannot be penetrated by a stand-alone fluorescent reporter (Kaljevic *et al.* 2021 Current Biology 31). A similarly tagged Bd0055 would therefore likely be confined to the outer, more accessible regions of the nucleoid, but we would have no idea to what extent this would be representative of the distribution of an untagged version in vivo. This is why we carried out orthogonal biochemical assays to confirm enrichment in the nucleoid.

6. The authors attempt to produce deletions in vivo to test if the Bd0055 is essential for “predatory and prey-independent” functions, without success. I don’t think that the absence of a result can be reported as evidence for anything. Instead they should do a conditional lethality assay to show that the gene is essential. In addition, how the revertant cell could regain the presence of the gene? What do the authors propose to be the function of Bd3044? Could it take over the functions of Bd0055?

Conditional lethality assays require knowledge of a (permissive) condition under which deletion is not lethal. We do not possess this knowledge for either Bd0055 or the *L. interrogans* histone. In the absence of a known mutant that can be exploited to establish conditional lethality, repeated attempts to delete a given gene without success remains the standard approach to determine whether the gene is essential. Genotyping >100 clones and establishing that not a single one of them contained a successful deletion is going above and beyond that standard. “Reversion” in this context does not mean that the gene is lost and then magically regained but rather reflects instances where the kanamycin cassette has integrated into the genome but a) not in the target locus, thus leaving the intended target intact or b) integrated into the target locus but leaving a second copy of the gene remained intact. This could be from a tandem duplication, or, more likely, as a result of merodiploidy (common in *B. bacteriovorus*), where the genome is present as >1 copy, creating a state where the histone can be deleted from one copy while the other one provides backup. It is standard terminology in the field – though admittedly confusing– to refer to these events as reversions. We have clarified this for the more general reader in the section “Bd0055 is essential for predatory and prey-independent growth”.

We currently do not know what the function of Bd3044 is; this is work in progress. Bd3044 expression is significantly lower than that of Bd0055 (see e.g. Fig 2b) so we do not think Bd3044 can act as a global organizer of chromatin. Evidently, Bd3044 does not readily take over the function of Bd0055. If that were the case, deletion of Bd0055 would not be lethal. We have added a sentence in the section “Bd0055 is essential for predatory and prey-independent growth” to make this explicit.

7. In the discussion, “it is unlikely that the nucleohistone filament is an artifact of excessive histonesin solution” is not a proof of the filament existing in vivo, and neither a proof that is not an artifact of crystallization.

We merely meant to indicate that it was not an overabundance of histones that ‘forced’ the crystal lattice into a nucleohistone filament. We have since obtained additional in-solution data that substantiate the profound differences between ‘canonical histone binding’ and the nucleohistone filament formed by Bd0055. We performed sedimentation velocity analyses demonstrating that Bd0055 saturates DNA at much higher concentrations than HTkA, consistent with the stoichiometry observed in the Bd0055 nucleohistone filament (described further below in response to reviewer #3 comment #9). These experiments are included in the revised version of the manuscript as new Fig. S8.

To avoid confusion we have deleted the sentence highlighted by the reviewer.

8. DNA binding by fluorescence was performed at a very low salt concentration of NaCl 10 mM. How this relates to the in vivo concentrations of ions? Would the binding still exist if salt concentrations are increased to physiological conditions? The same low salt concentration was used for incubation before the gel shifts.

We carried out additional experiments to show binding at a range of salt concentrations/pH conditions (included as new Fig S6 in the revised manuscript. These experiments suggest that the Bd0055 interaction with DNA is affected by salt and pH, as is the case for many DNA binding proteins, including archaeal/eukaryotic histones. To our knowledge, the salt/pH conditions that exist inside *Bdellovibrio* are not known. As proteins and other cellular components within the cytosol complex and sequester ions to various degrees, it is extremely difficult, if not impossible, to determine what buffer conditions particular molecules experience in cells. We used the buffer conditions we used because they gave us the tightest binding and the best opportunity to interrogate this interaction.

*9. They should make a comparison between the two different types of bacterial histones, singlets, and doublets, since there are three structures of doublet histone from *Aquifex aelicus* (Qui Yang, et al. (2003)), *Methanopyrus* (PDB 1F1E) and *Thermus thermophilus*. They should show the differences and similitudes with archaeal histones.*

In the current manuscript, we briefly point out that doublets in bacteria exist (which was already known from the structures highlighted by the reviewer and prior work by Alva and Lupas) and then focus on singlets, providing a like-for-like comparison with archaeal singlets (see e.g. Figures 1, 3, 4, 5). Doublets are interesting in their own right and we have been investigating their properties in a separate project. They have an evolutionary history that is distinct from singlets and many of them have lost the capacity to bind DNA. We believe that we cannot unravel this complexity as part of the current manuscript and including them here would confuse rather than illuminate. As for detailed

structural comparisons, we have often found these to be of little interest to a general audience, and have hence restrained ourselves. Of note, the reader can compare to their heart's content upon release of the PDB file, once the work has been formally accepted. All relevant files are, of course, available to the reviewers and have been included in the submission.

Minor comments

10. Several other histone-fold-related proteins have been reported, including histone-fold proteins such as TFIID, SAGA, and CENP-TSWX. Authors should comment on this.We have added this information to the Discussion, highlighting that there is precedent in histone fold domains having functions outside of DNA binding context (e.g. as dimerization domains) and that one should therefore keep an open mind when investigating the functional properties of other bacterial histones in the future.

11. Are the references related to the discussion in another file?

There was an unfortunate formatting error when generating the bibliography, which led to some references being omitted from the bibliography. This issue has been fixed.

12. The crystal structure of unbound Bd0055 seems to be a little bit over-refined. Rfree -Rwork =5.7%. But probably authors were planning to improve statistics for the rebuttal. Is there any reason why only a few waters were assigned in both crystals, even with such a high resolution? They should present the B-factors separated for DNA and protein, which would be useful for analysis.

The gap between Rfree and Rwork is a result of modelling the N-terminal residues for completeness, even though they are rather disordered. We decided to leave these amino acids in the final model, but here are the values (which we now state in the legend to Table S2):

- without N-terminal residues 2 and 3 (ala and glu, respectively) : Rwork-0.1932; Rfree-0.2259
- with N-terminal residues 2 and 3 (model in paper): Rwork-0.1878; Rfree-0.2335

We were conservative in the placement of water molecules, as one should be. B-factors are the same throughout the complex, with exception of the first two amino acids.

13. References should be corrected; the source of the tree is not correctly quoted as “were pruned from a phylogenetic tree based on the alignment of bacterial and archaeal histone-fold proteins (K. Stevens)”...including “(ref. 39)”.

This has been fixed.

14. Since there is an expression of other classical bacterial NAPs in B. bacteriovorus, what is the possibility that the function of Bd0055 is similar to SSB without the need to form the organelle they propose?

We presume that the reviewer refers to single-strand binding proteins here? *B. bacteriovorus* has an SSB homolog (Bd1582) with high structural conservation to other bacterial SSBs, so there is no obvious reason to suspect that Bd0055 would function as a single-stranded binding protein in this system, especially given that Bd0055 strongly binds dsDNA whereas well-characterized SSBs do not. One could investigate single-stranded DNA binding capacity of Bd0055 but we think this is of limited value: purified eukaryotic histones readily associate with single-stranded DNA, often with comparable affinity to double-stranded DNA (see e.g. Palter et al 1979 Cell (PID: 498278), Palter & Alberts 1979 JBC (PID: 500633)), so binding itself does not point to a key role in handling single-stranded DNA.

15. We consider this conclusion must change “histones are integral chromatin components across the tree of life and highlight organizational innovation in the domain Bacteria.” Some bacteria do not represent the whole kingdom.

As is evident from the remainder of the text, we did not mean to imply that histones are present throughout all of the bacteria. We have rephrased this sentence to read “histones **can be found as**[emphasis added] integral chromatin components across the tree of life...". We believe that this is accurate and will not mislead readers.

16. *Are the species shown in the bacterial species tree of Fig. 1d the only ones they found to have HLPs?. Otherwise, we suggest including a bacterial species tree annotated with the absence and presence of HLPs in the bacteria tree of life (similar to that in Fig 2a). From the bacterial species tree that is currently shown, apparently, only a few bacteria would have the full length of HLPs. Some of the bacteria seem to have only a short portion of the HF. Can the authors explain if they consider that these shorter peptides would have similar functions? What part of the HF those peptides would contain?*

Given the thousands of genomes we analyzed, visualizing all of them on a single tree and highlighting histone presence/absence is not practical. The tree in Figure 1D is, as highlighted in the figure, an abridged representation that reflects bacterial diversity. The full list of genomes with histones is provided in Table S1.

Regarding the second point, the shorter bars, rather than representing just a fraction of the histone fold, are full singlets, whereas the longer bars are doublets or, on occasion, singlets with additional domains/terminal extensions. We recognize that this was not obvious and have added further annotation to the figure and figure legend to make this clearer.

17. *In this manuscript, the authors propose that HLPs are used to organize bacterial chromosomes. This raises the question of at what point these bacteria obtained the genes in evolution. Some conclusions have already been proposed by the authors and others (Stevens et al., 2022; Alva and Lupas 2019). Based on the distribution of these proteins scattered in bacteria, do the authors hypothesize an origin from horizontal gene transfer? Otherwise, do these HLPs have a monophyletic origin? or are they a product of convergent evolution? Do you propose a pre-LUCA scenario? or LBCA? Could you comment on your work in the context of Moody et al 2022's work? I think it would be useful for the community.*

The evolutionary origin of bacterial histones is indeed a fascinating question. The phyletic distribution of bacterial singlet histones is very patchy and looks at first sight more like HGT than vertical descent. At the same time, all singlets share the four amino acid deletion in the middle of the $\alpha 2$ helix, which might support a monophyletic origin. We did not speculate about this in the original version of the manuscript because speculation is all we can ultimately offer. The way one would classically address these questions is via phylogenetics but, as we highlighted elsewhere (e.g. Stevens et al 2020 PNAS), prokaryotic histone genes are so short (<<100 amino acids) that the phylogenetic signal they bear is terrible and deep nodes in a phylogeny – the nodes we care about here – are invariably poorly supported. So even if we obtain a tree that points to a certain origin scenario, there will be no statistical support for that scenario.

Referee #2:

17Histones play a central role in chromatin remodeling, transcriptional regulation, and cellular control, for example. As such they are widely studied and very important proteins in biological research. Here, the authors present data suggesting that there are indeed histone like proteins involved in chromatin regulation in bacteria. These are very significant claims by the authors that here they support with a structural analysis of a protein which has a histone fold dimer, microscopy, DNA binding analyses, RNA Seq, and protein abundance analyses using proteomics methods.

Overall, the data is solid, but not wholly convincing that these proteins really behave like histones. To

a great extent this is a subjective consideration of the data. One could argue that the data they present is similar to many proteins that bind DNA and are involved in chromatin, but are not histones. Here, the histone fold data is the data that supports the main conclusion that leads to the title, for example. However, a major feature of histones in other organisms is the critical role that post translational modifications (acetylation, methylation, phosphorylation, etc.) play in their ability to regulate chromatin, for example.

This is the single major issue with this manuscript, especially for a journal like Nature. Are there regulatory post translational modifications on these proteins that play a critical role in the biology of these proteins in these organisms? Figure 1C suggests that this might be the case with conserved modifiable amino acids. To match the claims of the way this manuscript is currently written, and especially for a potential journal like Nature, this critical additional evidence is needed. Given the use of protein mass spectrometry by the authors to study to relative abundance of these proteins in cellular extracts, a detailed analysis of post translational modifications of these proteins in addition to the demonstration of the role such a modification may play in transcriptional regulation should be very doable.

Our conclusion that these are bona fide histone proteins is based on the only approach that can be used to reach that conclusion: homology. Whether post-translational modifications can be found has no bearing on whether that protein is a histone or not. Importantly, the ability of a eukaryotic histone to form a stable nucleosome is completely uncoupled from its modification state. Further, archaeal histones are rarely modified (reviewed in Stevens & Warnecke 2023 Seminars Cell Dev Biol), yet function perfectly fine as global organizers of DNA (see e.g. Mattioli et al 2017 Science). The notion that post-translational modifications are a defining feature of histones is therefore misguided.

That said, we agree with the reviewer that studying post-translational modifications is interesting and might ultimately add valuable clues regarding the potential for dynamic regulation of these histones. Note here that a prior dedicated PTM study of *L. interrogans* (Cao et al 2010 Cell Res 20:197) did not find evidence for modifications in its histone protein, which originally made us assign a low priority to this path of investigation. We have now re-analyzed our mass spectrometry data with a view to identifying post-translational modifications.

There are several constraints to this analysis, which bear pointing out. First, our mass spec protocol was not specifically geared towards identifying PTMs. We were principally interested in quantifying relative abundance. This means that we are blind to some modifications, e.g. phosphorylation, that require more targeted upstream processing for detection. Similarly, we did not include synthetic modified peptides that could have been used as standards to corroborate any inference of PTMs in native peptides. Second, some modifications that we can identify in principle from their shifted mass spectra, might be artifacts of sample processing. Notably, this includes lysine methylation, which has previously been linked to methanol extraction, which we use as part of the the nucleoid enrichment

protocol. These modification should therefore be interpreted with caution. Finally, and perhaps most importantly, the data in hand do not allow us to determine the stoichiometry of any given modification. This caveat is particularly relevant for high abundance proteins such as the histones analyzed here, as it is naturally easier to find a spurious modification in a protein present in high copy number.

With these caveats in mind, we focused on a set of modifications commonly associated with histones in eukaryotes [acetylation of lysine (K); ubiquitination of lysine/GlyGly (K); and methylation of lysine or arginine - methyl (K,R); dimethyl (K,R); trimethyl (K)]. Raw data were re-analysed using MaxQuant, focusing on sites where modifications could be inferred with high confidence by filteringto only include modifications with a posterior error probability (PEP) score below $1e-5$ and a localisation probability of 1. This is intentionally conservative to allow us to identify the most likely genuine modifications.

Across our datasets, we find no evidence for PTMs at highly conserved lysine (K10, K12, K16, K52) or arginine (R51) residues, so we do not think that high conservation reflects a role as a post-translational control switch. As summarized in Figure B below, we do find evidence for two modifications in Bd0055 (K18ac and K33me3) and two modifications in the *L. interrogans* histone A0A2H1XGH2 (K39me3, K39ub), at less well conserved lysines. Of note, we also detected K18ac in Bd0055 expressed from *E. coli*, perhaps suggesting that this residue is easily accessible and can therefore be acetylated as an off-target by different acetylases.

Figure B. Post-translational modifications detected in Bd0055 and A0A2H1XGH2 at PEP $1e-5$, summarized on the alignment between the two histones and presented in the context of amino acid conservation across bacteria. Mass spectra supporting individual PTMs are given below.

These results are intriguing but clearly insufficient to establish a role for PTMs in the physiology of bacterial histones. Rather, they should be seen as preliminary evidence that motivates future work, which should focus in particular on establishing whether any given modification is present at physiologically relevant stoichiometry. As such, we would advocate against inclusion of these data in the current manuscript, but are open to editorial guidance in this regard.

21Referee #3:

*The manuscript by Hocher et al. reports an intriguing mechanism of DNA compaction by bacterial (*B. bacteriovorus*) histone homolog Bd0055. The reported arrangement of nucleo-protein filament is completely unique among known histone-mediated genome compaction mechanisms, wherein DNA is totally protected from solvent by the associated protein. Potentially this is a rather illuminating and instructive discovery, demonstrating bacterial evolutionary innovation on the grand scale and highlighting unique opportunities for gene regulation, replication control and stress tolerance presented by solvent-excluded (inert) genomic DNA.*

Unfortunately, as presented in the manuscript, the data and its analysis do not allow to fully evaluate the validity of claims outlined in the report. Comments below highlight specific deficiencies and/or concerns.

1. Since 8fvx and 8fw7 solutions are on hold at rscb.org, evaluation of crystallographic solutions of recombination Bb0055 in apo-form and bound to short (35 bp) DNA fragment is limited to authors-generated figures. The quality of solutions cannot be ascertained due to the omission of the pdb report.

We are sorry that the reviewer was not given access to all the underlying raw data. We obviously agree that *all* raw data (including structural and proteomic data) should be available for scrutiny by the reviewers. We had forwarded the PDB report to the editor who failed to share it with the reviewers. We have made sure that the PDB reports and structures have been included in this submission (see below for a link to the proteomics data).

1b. Visually the solution of the apo-form (Fig. 3) is similar to the higher resolution structure of Bb0055 (8cmp, doi.org/10.1101/2023.02.26.530074); however, this is neither surprising, nor an independent confirmation (given that both solutions were obtained by molecular replacement). Fig 5 illustrates details of Bb0055-DNA binding in combined stick-cartoon representation, leaving opaque the composition and architecture of the actually solved structure.

Comparing two pictures from pre-print PDFs is obviously not a good way to compare structures and the reviewers should have been given access to the relevant raw data to explore the structure. That said, we are not sure what to make of the statement that ‘*visually the solution of the apo-form (Fig. 3) is similar to the higher resolution structure of Bb0055 (8cmp, doi.org/10.1101/2023.02.26.530074); however, this is neither surprising, nor is it an independent confirmation (given that both solutions were obtained by molecular replacement)*’. One would indeed hope that the later structure (8cmp, which is also on hold on the PDB) is the same as our apo structure, since they are the same protein. Incidentally, different search models for phasing the two structures were used, and as such it is a good independent confirmation of our structure (not that that would have been needed). Molecular replacement is an acknowledged technique that has been used for many decades to solve protein structures. In this case, the search model accounted for 100 % of the amino acids which is as good as it gets. We hope that the availability of the PDB file (now uploaded) will enable the reviewer to make more detailed structural comparisons.

2. Proteomics raw data (PRD039405) isn't available for re-analysis (no such entry at PRIDE db);

The data were available through reviewer tokens, but unfortunately do not show up when entering the ID at the PRIDE homepage. The following link on ProteomeXchange (<http://proteomecentral.proteomexchange.org/cgi/GetDataset?ID=PX039405>) does show the data

23as reserved on PRIDE. Reviewers can access the data using the following details:

Go to <https://www.ebi.ac.uk/pride/login>

Username: reviewer_pxd039405@ebi.ac.uk

Password: 99KlcTnQ

3. based on the description the analysis was performed perfunctorily (e.g. as evidenced by inclusion of arbitrary “constant” and “variable” modifications into the search, instead of discovery of relevant modifications actually present in the sample via “open search” inference).The principal purpose of our proteomics analysis (and therefore upstream workflow) was to determine whether histones are highly abundant rather than identify PTMs. Within the limitations of the data generated through this workflow, we have now looked for histone post-translational modifications in both *B. bacteriovorus* and *L. interrogans*. Please see our response to reviewer #2 for the results of this analysis and its caveats.

4. *Description of the data in the text doesn't correspond to the figures referenced therein. For example, p. 9 lane 15 reports A0A2H1XGH2 as ranked 4 of 1502 quantified proteins (Figs 6a, S11), whereas none of the referenced Figs contains any ranked lists.*

Old Figure S11 (now Figure S14) shows ranked protein abundances in *L. interrogans*. The histone ranks 4 of 1502 quantified proteins in a prior study by Schmidt et al, which is summarized in Figure S14a. This was ambiguous in the prior version of the manuscript and has now been clarified.

5. *Moreover, Fig 11a has ranking graph going up to about 1500 (referencing separate work by Schmidt et al, 2011), Fig 11b reporting data from manuscript under review, extends the ranking graph to over 2500 (in contrast to text claiming quantitation of 1502 proteins).*

See above. The figure of 1502 proteins refers to the study of Schmidt et al. The ranks in old Figure S11b (now Figure S14b) go beyond 2500 for the very simple reason that we quantified more proteins than Schmidt et al.

6. *Report of Bb0055 "high abundance" in the cell plays a substantial role in the overall narrative of the manuscript, however, the data was obtained in the most unsophisticated way (use of one-cartridge kit for modification/digestion/purification, lack of internal control (such as spiking of the sample with known quantities of Bb0055 protein and/or peptides), analysis was carried out at the level lacking appropriate sophistication (see above and bacterial proteome quantitation reports (e.g. doi/full/10.15252/msb.20209536). Results of quantitation are reported in the way opaque for general reader (MS intensities).*

The kit and process we use are perfectly adequate for the questions we wanted answered. Added sophistication is simply not required here, in particular given that our relative quantitation for *L. interrogans* correlates very well with the absolute quantitation by Schmidt *et al.*, which the reviewer considers to be of superior quality (Spearman's rho = 0.66, Figure C below, now also included in the manuscript as Figure S14c). Our workflow is based on the work of the Mann lab (see Müller et al 2020 Nature 582), an undisputed leader in the proteomics field. We think that the reviewer's comments here are not warranted.

Figure C. Comparison between absolute protein abundance values reported in Schmidt *et al.* (2011) and relative abundance values in our study.

7. More transparently and acceptably protein abundances ought to be reported as fractions of the total proteome mass, or, most appropriately for general reader, in molecules per cell. The latter was utilized in Fig. S11a, citing Schmidt *et al.*, 2011.

First, our data correlate highly with that of Schmidt *et al.* (see above), testifying to the quality of our data. Second, we can report the data in a different way, e.g. as a fraction of proteome mass but that changes nothing. It just adds the same denominator to all abundance values. We cannot report molecules per cell as we did not seek absolute quantification and therefore did not include spike-in standards (and cell counting) required for that purpose.

8. It is not clear why the authors used these data together with their own without discussion: reported in S11a abundance for “one of the most abundant proteins” in *L. interrogans* appears to be slightly over 14 molecules per cell. In bacterial proteomes highest abundance of proteins is in tens of thousands per cell, not in low teens. The authors need to repeat their LFQ experiments, using spike of known quantity of Bb0055 added to pre-measured amount of cell material for conversion of intensities into molecules/cell. So far the only quantitation given in the manuscript, ~14 molecules/cell, allows for the formation of a nucleo-histone filament ~20 bp long, negligible compared to the size of bacterial genome.

This comment derives from an unfortunate mistake in labelling the y axis of Figure S11a (now Figure S14a). The data are plotted on a log₂ scale, so it is not 14, but rather 2¹⁴ (>16,000) molecules per cell!

To reiterate, absolute quantification in the manner described by the reviewer is absolutely **not** required to support the conclusions we draw.

9. MD simulations carried out with the model of Bb0055 tetramer (p.7) cannot be used to judge the stability of said tetramer in the cell, not even the stability of the same at the simulation conditions (300K, water, K+, Amber18 force field). All that can be reliably concluded from this simulation is that the model as built (by docking or superimposing (the authors use different descriptors and provide no details for this operation) the structure of Bb0055 monomer into the experimental structure (5t5k) of archeal “hypernucleosome”) is unstable in MD simulation. It is a common occurrence with docking/superimposed models, which usually reflects on the quality of the model or its refinement, rather than on the physico-chemical properties of the actual complex. Multimerization of the Bb0055

can be assayed directly by CG-MALS, HM-MS (high mass or “native” protein MS), XL-MS, and a number of other experimental techniques.

There are two distinct points here. First, we agree that MD comparisons of observed and homology-modelled structures cannot offer conclusive support for a given molecular behaviour. We use MD here judiciously, arguing solely that the simulated behaviour, i.e. that Bd0055 does not form stable hypernucleosomal complexes, is consistent with multiple orthogonal lines of evidence (FP, FRET, EMSA).

Second, the reviewer suggests alternative experimental techniques to support our claims of different multimerization dynamics of Bd0055 on DNA. In recognition of this, and to provide further evidence for the profound differences between Bd0055 and HTkA in this regard, we now include new analytical ultracentrifugation (AUC) experiments that show that Bd0055 assembles onto DNA to form a complex with a much larger sedimentation coefficient than HTkA (summarized in Figure D below, see revised manuscript for further details). This experiment also shows that it takes much more Bd0055 to fully saturate a particular piece of DNA, consistent with our model and crystal structure.Figure D. Analytical ultracentrifugation of WT and mutant Bd0055 and HTkA with 147bp of dsDNA.10. Cellular context and therefore biological relevance of the reported histone-like properties of Bb0055 are largely opaque or missing. Lacking any other absolute quantitation, 14 molecules of this protein in the cell are unlikely to make a big impact on cellular function by completely isolating ~20 bp of DNA from solution. ChIP-Seq or at least a meaningful proteomics survey would go a long way toward elucidation of Bb0055 biological role and the potential scale of its effect. If the assessment of Bb0055 high abundance holds up, then cryo-ET of the filaments formed by the same *in vivo* would greatly advance the argument for the existence of nucleo-histone filaments in bacteria.

See above regarding *in vivo* testing and “14 molecules per cell”. We agree that cryo-ET will be useful in the future to further elucidate how Bb0055 and other bacterial histones structure the nucleoid *in vivo*.

Decision Letter, first revision:

Message: Our ref: NMICROBIOL-23051049A

14th August 2023

Dear Dr. Warnecke,

Thank you very much for your patience as we've prepared the guidelines for final submission of your Nature Microbiology manuscript, "Histones with an unconventional DNA binding mode are major chromatin constituents in the bacterium *Bdellovibrio bacteriovorus*" (NMICROBIOL-23051049A). I apologise for the time taken to get this back to you and hope that the attached documents are clear. Please carefully follow the step-by-step instructions provided in the attached file, and add a response in each row of the table to indicate the changes that you have made. Please also check and comment on any additional marked-up edits we have proposed within the text. Ensuring that each point is addressed will help to ensure that your revised manuscript can be swiftly handed over to our production team.

We would like to start working on your revised paper, with all of the requested files and forms, as soon as possible (preferably within two weeks). Please get in contact with us if you anticipate delays, or if you have any questions related to the checklist and associated documents.

If you have not done so already, please alert us to any related manuscripts from your group that are under consideration or in press at other journals, or are being written up for submission to other journals (see: <https://www.nature.com/nature-research/editorial->

1policies/plagiarism#policy-on-duplicate-publication for details).

In recognition of the time and expertise our reviewers provide to Nature Microbiology's editorial process, we would like to formally acknowledge their contribution to the external peer review of your manuscript entitled "Histones with an unconventional DNA binding mode are major chromatin constituents in the bacterium *Bdellovibrio bacteriovorus*". For those reviewers who give their assent, we will be publishing their names alongside the published article.

Nature Microbiology offers a Transparent Peer Review option for new original research manuscripts submitted after December 1st, 2019. As part of this initiative, we encourage our authors to support increased transparency into the peer review process by agreeing to have the reviewer comments, author rebuttal letters, and editorial decision letters published as a Supplementary item. When you submit your final files please clearly state in your cover letter whether or not you would like to participate in this initiative. Please note that failure to state your preference will result in delays in accepting your manuscript for publication.

Cover suggestions

As you prepare your final files we encourage you to consider whether you have any images or illustrations that may be appropriate for use on the cover of Nature Microbiology.

Nature Microbiology has now transitioned to a unified Rights Collection system which will allow our Author Services team to quickly and easily collect the rights and permissions required to publish your work. Approximately 10 days after your paper is formally accepted, you will receive an email in providing you with a link to complete the grant of rights. If your paper is eligible for Open Access, our Author Services team will also be in touch regarding any additional information that may be required to arrange payment for your article.

Please note that you will not receive your proofs until the publishing agreement has been

received through our system.

Please note that *Nature Microbiology* is a Transformative Journal (TJ). Authors may publish their research with us through the traditional subscription access route or make their paper immediately open access through payment of an article-processing charge (APC). Authors will not be required to make a final decision about access to their article until it has been accepted. [Find out more about Transformative Journals](https://www.springernature.com/gp/open-research/transformative-journals)

Authors may need to take specific actions to achieve [compliance with funder and institutional open access mandates](https://www.springernature.com/gp/open-research/funding/policy-compliance-faqs). If your research is supported by a funder that requires immediate open access (e.g. according to [Plan S principles](https://www.springernature.com/gp/open-research/plan-s-compliance)) then you should select the gold OA route, and we will direct you to the compliant route where possible. For authors selecting the subscription publication route, the journal's standard licensing terms will need to be accepted, including [self-archiving policies](https://www.nature.com/nature-portfolio/editorial-policies/self-archiving-and-license-to-publish). Those licensing terms will supersede any other terms that the author or any third party may assert apply to any version of the manuscript.

[redacted]

[redacted]

Reviewer #1:

Remarks to the Author:

This time the authors have addressed all my concerns. I believe the paper is now acceptable.

Final Decision Letter:

Message: 8th September 2023

Dear Toby,

Thank you very much for your time this morning. I am really pleased to accept your Article "Histones with an unconventional DNA binding mode are major chromatin constituents in the bacterium *Bdellovibrio bacteriovorus*" for publication in Nature Microbiology. Thank you for having chosen to submit your work to us and many congratulations.

Acceptance of your manuscript is conditional on all authors' agreement with our publication policies (see <https://www.nature.com/nmicrobiol/editorial-policies>). In particular your manuscript must not be published elsewhere and there must be no announcement of the work to any media outlet until the publication date (the day on which it is uploaded onto our website).

Please note that *Nature Microbiology* is a Transformative Journal (TJ). Authors may publish their research with us through the traditional subscription access route or make their paper immediately open access through payment of an article-processing charge (APC). Authors will not be required to make a final decision about access to their article until it has been accepted. [Find out more about Transformative Journals](https://www.springernature.com/gp/open-research/transformative-journals)

Authors may need to take specific actions to achieve [compliance with funder and institutional open access](https://www.springernature.com/gp/open-research/funding/policy-compliance-faqs)

4mandates. If your research is supported by a funder that requires immediate open access (e.g. according to [Plan S principles](https://www.springernature.com/gp/open-research/plan-s-compliance)) then you should select the gold OA route, and we will direct you to the compliant route where possible. For authors selecting the subscription publication route, the journal's standard licensing terms will need to be accepted, including [self-archiving policies](https://www.nature.com/nature-portfolio/editorial-policies/self-archiving-and-license-to-publish). Those licensing terms will supersede any other terms that the author or any third party may assert apply to any version of the manuscript.

With kind regards and have a good weekend,